# Optical manipulation of sphingolipid biosynthesis using photoswitchable ceramides

**Matthijs Kol[1†], Ben Williams[2†], Henry Toombs-Ruane[2], Henri G Franquelim[3], Sergei Korneev[1], Christian Schroeer[1], Petra Schwille[3], Dirk Trauner[4]\*, Joost CM Holthuis[1]\*, James A Frank[5]\***

[1]Department of Biology/Chemistry, University of Osnabrück, Osnabrück, Germany; [2]Department of Chemistry, Ludwig Maximilians University Munich, Munich, Germany; [3]Department of Cellular and Molecular Biophysics, Max Planck Institute of Biochemistry, Martinsried, Germany; [4]Department of Chemistry, New York University, New York, United States; [5]The Vollum Institute, Oregon Health and Science University, Portland, United States

**\*For correspondence:**
dirktrauner@nyu.edu (DT);
Joost.Holthuis@Biologie.Uni-Osnabrueck.DE (JCMH);
frankja@ohsu.edu (JAF)

[†]These authors contributed equally to this work

**Competing interests:** The authors declare that no competing interests exist.

**Abstract** Ceramides are central intermediates of sphingolipid metabolism that also function as potent messengers in stress signaling and apoptosis. Progress in understanding how ceramides execute their biological roles is hampered by a lack of methods to manipulate their cellular levels and metabolic fate with appropriate spatiotemporal precision. Here, we report on clickable, azobenzene-containing ceramides, caCers, as photoswitchable metabolic substrates to exert optical control over sphingolipid production in cells. Combining atomic force microscopy on model bilayers with metabolic tracing studies in cells, we demonstrate that light-induced alterations in the lateral packing of caCers lead to marked differences in their metabolic conversion by sphingomyelin synthase and glucosylceramide synthase. These changes in metabolic rates are instant and reversible over several cycles of photoswitching. Our findings disclose new opportunities to probe the causal roles of ceramides and their metabolic derivatives in a wide array of sphingolipid-dependent cellular processes with the spatiotemporal precision of light.
DOI: https://doi.org/10.7554/eLife.43230.001

## Introduction

Sphingolipids are unusually versatile membrane components in eukaryotic cells that contribute to mechanical stability, cell signaling and molecular sorting (*Holthuis et al., 2001*). They derive from the addition of various polar head groups to ceramide, a hydrophobic molecule containing saturated or *trans*-unsaturated acyl chains linked to a serine backbone (*Wegner et al., 2016*). The enzymes responsible for sphingolipid production and turnover comprise a complex metabolic network that gives rise to numerous bioactive molecules. Intermediates of sphingolipid metabolism, notably sphingosine, ceramide and their phosphorylated derivatives, influence a multitude of physiological processes, including cell proliferation, cell death, migration, stress adaptation, immune responses and angiogenesis (*Hannun and Obeid, 2018*). Consequently, imbalances in sphingolipid metabolism are linked to major human diseases (*Ogretmen, 2018*).

Sphingomyelin (SM) and glycosphingolipids (GSLs) are the main sphingolipid classes in mammals (*Koval and Pagano, 1991*; *D'Angelo et al., 2013*). SM forms a concentration gradient along the secretory pathway and influences cellular cholesterol homeostasis to sustain vital physical properties of the plasma membrane, where SM and Chol are enriched (*Slotte, 2013*). The hydrolysis of plasma membrane SM by sphingomyelinases generates ceramide, which mediates several stress responses

including cell cycle arrest, senescence and apoptosis (*Adada et al., 2016*). SM biosynthesis is mediated by SM synthase (SMS), an enzyme catalyzing the transfer of phosphocholine from phosphatidylcholine (PC) onto ceramide, yielding SM and diacylglycerol (DAG). Mammals contain two SMS isoforms, namely SMS1, responsible for bulk production of SM in the Golgi lumen, and SMS2, serving a role in regenerating SM from ceramides released by sphingomyelinases on the cell surface (*Huitema et al., 2004*; *Yamaoka et al., 2004*). Besides their biological potential as cross-regulators of the pro-apoptotic factor ceramide and mitogenic factor DAG, studies in mice revealed roles for SMS1 and SMS2 in inflammation, atherosclerosis and diabetes (*Mitsutake et al., 2011*; *Li et al., 2011*; *Liu et al., 2009*). Conversely, GSLs are synthesized by the sequential action of Golgi-resident glycosyltransferases. These enzymes conjugate ceramide to a specific carbohydrate from a sugar nucleotide (e.g. UDP-glucose, UDP-galactose), or onto a ceramide-conjugated carbohydrate chain, giving rise to a large variety of structurally and functionally divergent compounds (*D'Angelo et al., 2013*). While GSLs are dispensable for cell survival, they are key to cell-cell communication and collectively required for mammalian development (*D'Angelo et al., 2013*; *Yamashita, 2000*).

Progress in understanding how sphingolipids exert their multitude of tasks is hindered by a lack of suitable methods to probe lipid function (*Honigmann and Nadler, 2018*). Manipulation of cellular lipid pools by altering expression of metabolic enzymes is a slow process, allowing cells to mount an adaptive response that dampens functional impact. The fact that lipid metabolic networks are highly interconnected also makes it difficult to prove that the direct product of a specific enzyme is the actual effector. Chemical dimerizers and optogenetic approaches allow researchers to manipulate lipid levels more rapidly, but their application often requires extensive protein engineering and is largely restricted to soluble lipid-metabolic enzymes or transfer proteins (*Jain et al., 2017*). Another method to rapidly increase lipid supply involves the use of caged lipids. These compounds carry a photo-labile protecting group that blocks their biological activity until the active lipid is released with a flash of light (*Kim et al., 2016*; *Höglinger et al., 2014*; *Höglinger et al., 2015*). Alongside the recent development of site-directed variants of caged sphingosine, it has become possible to confine release of this bioactive lipid to specific organelles (*Feng et al., 2018*). However, once triggered, the activity cannot be switched off and the decay of the lipid signal depends on its metabolic conversion. In this respect, photoswitchable azobenzene-containing lipids hold great promise by allowing the translation of optical stimuli into a reversible cellular response (*Frank et al., 2015*; *Frank et al., 2016a*; *Frank et al., 2016b*). In previous studies, we demonstrated the potency of azobenzene-containing fatty acids as photoswitchable agonists of the vanilloid receptor 1 (*Frank et al., 2015*), and found that photoswitchable DAGs enable light-mediated oscillations in protein kinase C activation (*Frank et al., 2016a*).

We recently synthesized a set of azobenzene-containing ceramides (ACes) to exert optical control over the curvature of the *N*-acyl chain. All three ACes displayed a light-dependent cycling between liquid-ordered ($L_o$) and liquid-disordered ($L_d$) domains when incorporated into phase-separated supported lipid bilayers (SLBs), thus enabling dynamic manipulation of membrane structure and fluidity (*Frank et al., 2016b*). Whether such light-mediated restructuring of membranes can also be implemented in living systems remained unclear. Moreover, whether azo-ceramides can be exploited to reversibly manipulate sphingolipid metabolism or function remained to be established. Here, we introduce a new generation of photoswitchable ceramides containing an azobenzene in the sphingoid base or *N*-acyl chain, along with a clickable alkyne group for increased metabolite detection sensitivity. Using both synthetic bilayers and living cells as experimental models, we demonstrate the potential of these compounds as photoswitchable substrates to place sphingolipid biosynthesis under the dynamic control of light.

## Results

### Chemical synthesis of clickable and photoswitchable ceramides

We designed and synthesized four **c**lickable and **a**zobenzene-containing **cer**amide analogues, **caCer-1–4** (*Figure 1a,b*, *Supplementary file 1*). Each caCer has an *N*-acyl chain possessing a terminal alkyne for *click*-derivatization with a fluorophore reporter. **caCer-1** is a clickable variant of **ACe-1** on which we reported earlier (*Frank et al., 2016b*) (*Figure 1c*), and carries the azobenzene photoswitch in its *N*-acyl chain with the diazene group mimicking a $\Delta^9$-double bond. **caCer-2** has a shorter *N*-acyl

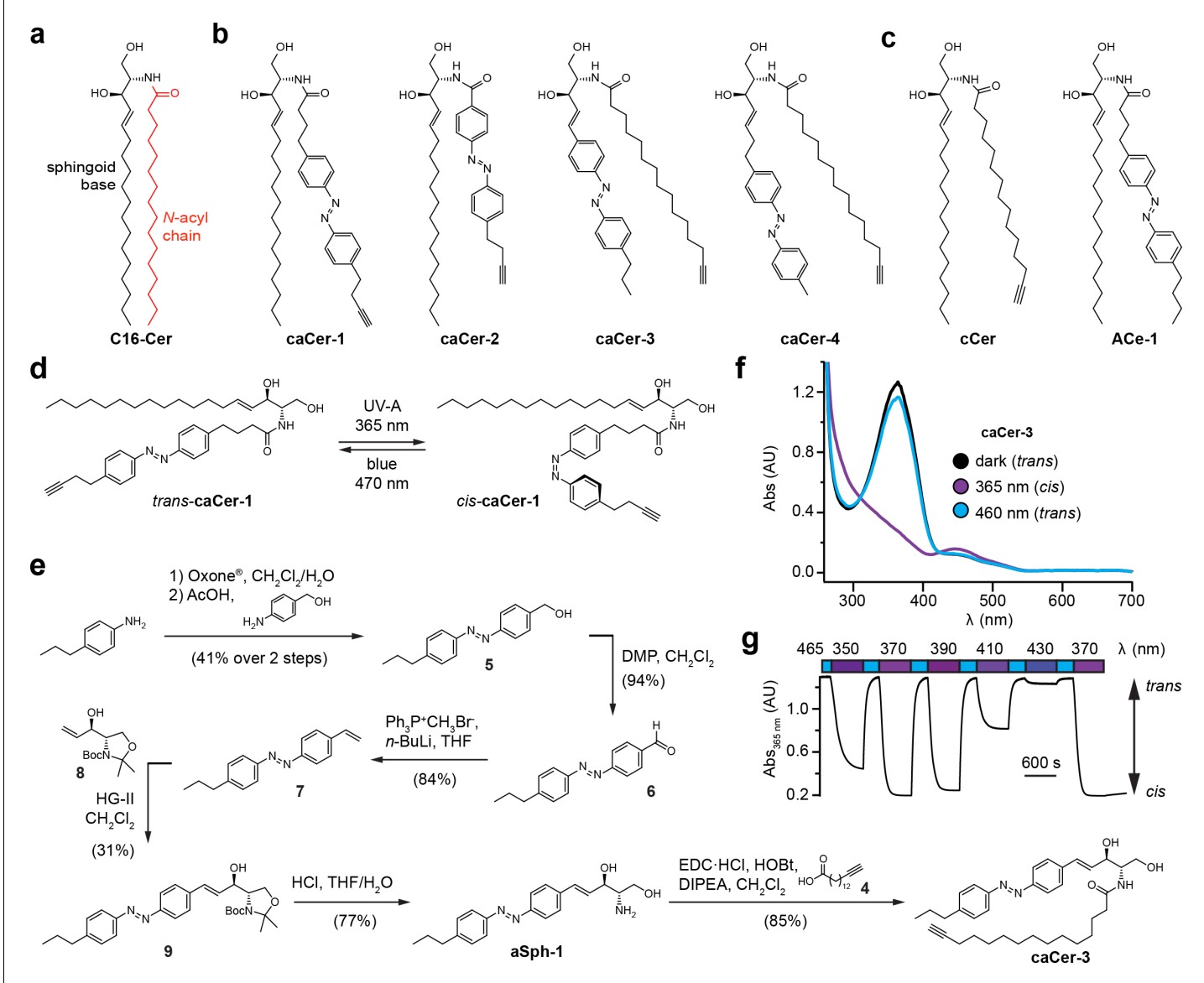

**Figure 1.** Design and synthesis of clickable and photoswitchable ceramides. (**a**) Chemical structure of C16:0-Cer, with color-coded sphingoid base (black) and *N*-acyl chain (red). (**b**) Chemical structures of clickable and photoswitchable ceramides **caCer-1–4**. (**c**) Chemical structures of clickable ceramide **cCer** and photoswitchable ceramide **ACe-1**. (**d**) **caCer-1** undergoes reversible isomerization between its *cis*- and *trans*-configuration with UV-A (365 nm) and blue (470 nm) light, respectively. (**e**) Chemical synthesis of the photoswitchable sphingoid base **aSph-1**, which was *N*-acylated with a terminal alkyne-functionalized C15 fatty acid to afford **caCer-3**. (**f**) UV-Vis spectra of the red-shifted variant **caCer-3** (50 μM in DMSO) in its dark-adapted (*trans*, black), UV-A-irradiated (*cis*, violet) and blue-irradiated (*trans*, blue) states. (**g**) **caCer-3** (50 μM in DMSO) undergoes isomerization to its *cis*-configuration with UV-A light (350–390 nm), and this effect is completely reversed with blue light (465 nm). Photoswitching was monitored by measuring the absorbance at 365 nm.

DOI: https://doi.org/10.7554/eLife.43230.002

The following figure supplements are available for figure 1:

**Figure supplement 1.** Chemical synthesis of caCer-1.
DOI: https://doi.org/10.7554/eLife.43230.003
**Figure supplement 2.** Chemical synthesis of caCer-2.
DOI: https://doi.org/10.7554/eLife.43230.004
**Figure supplement 3.** UV-Vis spectra of caCer-1, caCer-2 and Cer-4.
DOI: https://doi.org/10.7554/eLife.43230.005
**Figure supplement 4.** Chemical synthesis of caCer-4.
DOI: https://doi.org/10.7554/eLife.43230.006

chain with an azobenzene whose diazene group mimics a $\Delta^6$-double bond. These two analogues were synthesized by first converting the commercially available aldehyde **1** to terminal alkyne **2** with the Ohira-Bestmann reagent (*Ohira, 1989*; *Müller et al., 1996*). A tin dichloride-mediated reduction then afforded aniline **3**, which was subjected to the Baeyer-Mills reaction (*Baeyer, 1874*; *Mills, 1895*). Using different nitroso compounds, the functionalized fatty acids **cFAAzo-4** (*Figure 1— figure supplement 1*) and **cFAAzo-1** (*Figure 1—figure supplement 2*) could be prepared. Amidation of **cFAAzo-4** and **cFAAzo-1** with D-*erythro*-sphingosine afforded **caCer-1** and **caCer-2**, respectively. UV-Vis spectroscopy revealed that both compounds behaved as standard azobenzenes (*Figure 1—figure supplement 3a,b*). In the dark, they existed in their thermally stable *trans*-configuration, and could be isomerized to *cis* using UV-A (350–390 nm) illumination. The *trans*-isomer could be regenerated using blue (470 nm) light (*Figure 1d*).

caCer-3 and **caCer-4** were designed around two distinct azobenzene-containing sphingoid bases, **aSph-1** and **aSph-2**, which possessed either a two- or four-carbon linker between the head group and the azobenzene, respectively (*Figure 1e*, *Figure 1—figure supplement 4*). The styrene **aSph-1** was synthesized from 4-propylaniline, which was converted by Baeyer-Mills reaction to the hydroxy-methyl azobenzene **5** (*Figure 1e*). Sequential oxidation to aldehyde **6** and Wittig reaction afforded olefin **7**. Cross-metathesis (*Rai and Basu, 2004*; *Peters et al., 2007*) of **7** with the L-serine-derived allyl alcohol **8** (*Garner, 1984*; *Garner and Park, 1992*) followed by deprotection afforded **aSph-1**. Acylation with the C15 terminal alkyne-tagged fatty acid **4** (*Seike et al., 2006*) (*Figure 1—figure supplement 4a*) afforded **caCer-3** (*Figure 1e*). **caCer-4** was prepared in an analogous fashion from **aSph-2** (*Figure 1—figure supplement 4b*). UV-Vis spectroscopy revealed that the styrene **caCer-3** possessed a slightly red-shifted absorption spectrum, with a $\lambda_{max}$ at 365 nm (*Figure 1f*) compared to about 335 nm for the rest of the caCers (*Figure 1—figure supplement 3*). To determine the optimal photoswitching wavelength to generate *cis*-**caCer-3**, we continuously monitored the absorbance at 365 nm in solution and performed a wavelength scan between 350 to 410 nm. We found that the *trans* to *cis* isomerization of **caCer-3** was most efficient at 370 nm, similar to the rest of the caCers and ACes (*Frank et al., 2016b*) (*Figure 1g*). **caCer-4** possessed similar absorption spectra as **caCer-1** and **caCer-2** (*Figure 1—figure supplement 3c*).

## caCers enable optical control of ordered lipid domains in supported bilayers

We previously reported that *N*-acyl-azobenzene ceramides analogous to **caCer-1** and **caCer-2**, but lacking the alkyne functionality, can be used to optically manipulate the structure of phase-separated lipid domains in SLBs (*Frank et al., 2016b*). However, whether ceramides with azobenzene-containing sphingoid bases display a similar behavior in such lipid environments was still in question. We therefore performed confocal fluorescence imaging and scanning atomic force microscopy (AFM) on SLBs containing a quaternary lipid mixture of 1,2-*O*-dioleoyl-*sn*-glycero-3-*O*-phosphocholine (DOPC), cholesterol, SM (C18:0) and **caCer-3** or **caCer-4** at a molar ratio of 10:6.7:5:5 DOPC:cholesterol:SM:caCer (*Figure 2*). Each lipid mixture was doped with 0.1 mol% ATTO-655-DOPE to allow visualization of domain segregation in the SLB using confocal microscopy through the preferential localization of the dye in liquid-disordered ($L_d$) membrane regions (*Frank et al., 2016b*). Confocal fluorescence imaging of SLBs from lipid mixtures containing the dark-adapted *trans*-caCers revealed a clear phase separation, with liquid-ordered ($L_o$) domains appearing as dark regions (*Figure 2a*).

Imaging of the SLBs by AFM confirmed the formation of $L_o$ domains, which rested approximately 1 nm above the $L_d$ phase (*Figure 2b*). UV-A-induced isomerization of dark-adapted caCers to *cis* led to a rapid fluidification of the $L_o$ domains, as indicated by the appearance of many small liquid-disordered ($L_d$) 'lakes' within the $L_o$ domains, alongside an increase in the $L_d/L_o$ area ratio. During equilibration, these lakes laterally diffused toward the $L_d$ phase or coalesced into larger lakes in an effort to minimize line tension. On isomerization of **caCer**s to *trans* by blue light, the $L_d$ lakes shrunk in size while the surrounding $L_o$ areas expanded, essentially occupying the same total area as in the original dark-adapted state. These light-induced effects on lipid domains were observed for both **caCer-3** (*Figure 2b*, top) and **caCer-4** (*Figure 2b*, bottom) and could be repeated over multiple cycles of UV-A and blue light illumination (*Figure 2c*, *Figure 2—video 1* and *2*). This indicates that, while *trans*-isomers of **caCer-3** and **caCer-4** preferentially localize to cholesterol- and SM-rich $L_o$ domains, their *cis*-isomers favor association with more loosely packed lipids like DOPC. Hence, ceramides

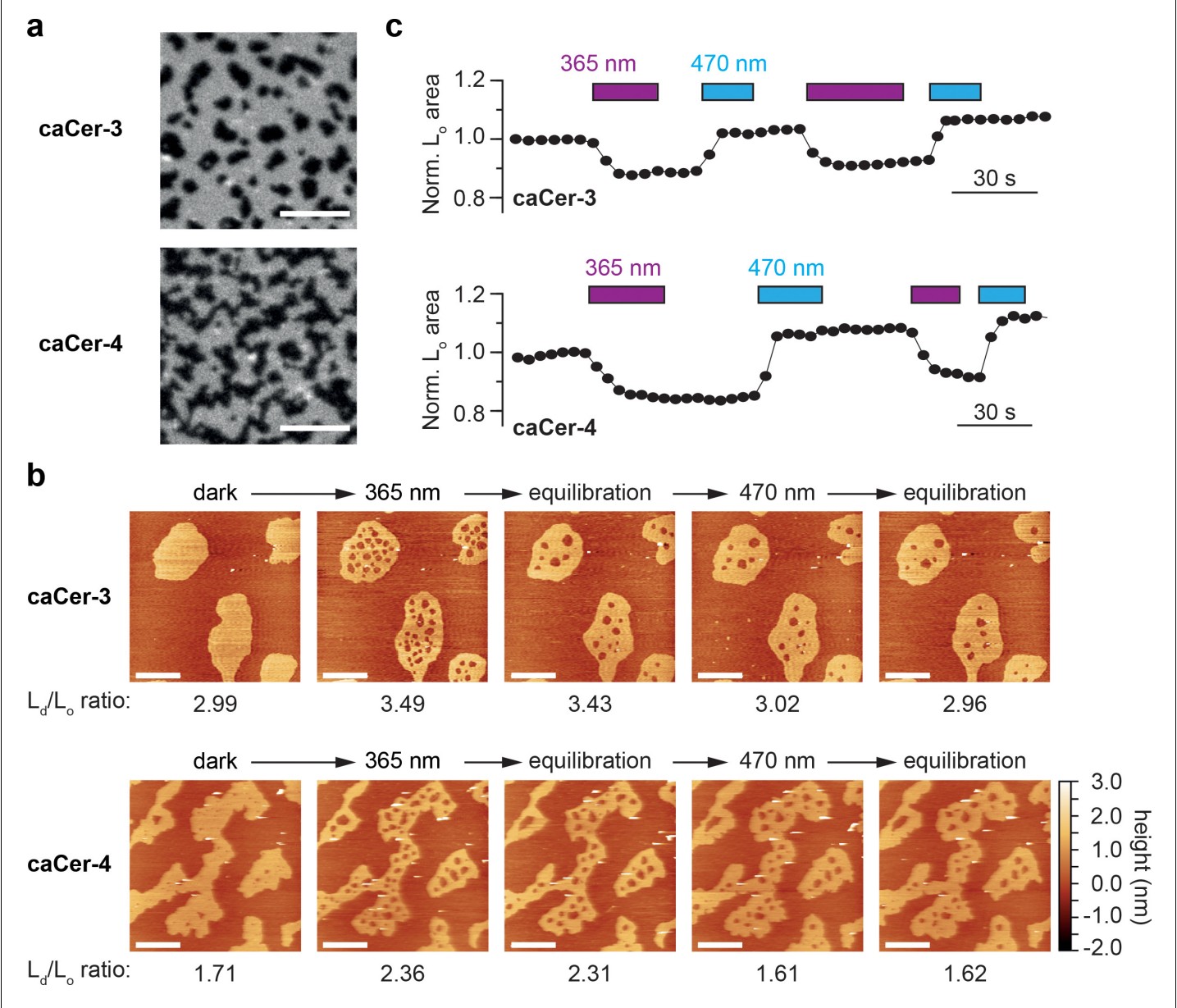

**Figure 2.** Photo-isomerization of caCers affects membrane fluidity and lipid domain structure in supported lipid bilayers. (**a**) Confocal fluorescence microscopy of SLBs containing a quaternary mixture of DOPC:cholesterol:SM:**caCer** (10:6.7:5:5 mol ratio) and 0.1 mol% ATTO655-DOPE. Images of SLBs prepared with dark-adapted *trans*-**caCer-3** and *trans*-**caCer-4** revealed phase separation, with taller liquid-ordered ($L_o$) domains appearing as dark regions in a liquid-disordered ($L_d$) phase. Scale bars, 10 μm. (**b**) Atomic force microscopy of SLBs prepared as in (**a**). Isomerization of **caCer-3** (top) and **caCer-4** (bottom) to *cis* with UV-A light (365 nm) resulted in a fluidification inside the $L_o$ domains, as indicated by the appearance of small fluid $L_d$ lakes and an increased $L_d/L_o$ area ratio. This effect was reversed on isomerization back to *trans* with blue light (470 nm), marked by a drop in the $L_d/L_o$ area ratio. Scale bars, 2 μm. (**c**) Time-course plotting the normalized $L_o$ area over multiple 365/470 nm irradiation cycles for **caCer-3** (top) and **caCer-4** (bottom).

DOI: https://doi.org/10.7554/eLife.43230.007
The following videos are available for figure 2:
**Figure 2—video 1.** Reversible remodeling of lipid domains by caCer-3.
DOI: https://doi.org/10.7554/eLife.43230.008
**Figure 2—video 2.** Reversible remodeling of lipid domains by caCer-4.
DOI: https://doi.org/10.7554/eLife.43230.009

carrying the azobenzene in their sphingoid backbone display similar light-dependent preferences for $L_o$ and $L_d$ domains as those reported for *N*-acyl-azobenzene ceramides (*Frank et al., 2016b*).

## caCers are light-sensitive substrates of SM synthase

To address whether the photoswitchable properties of caCers can also be exploited in native cellular membranes, we first monitored their metabolic conversion into SM by human SMS2 heterologously expressed in yeast. To this end, caCers were kept in the dark or pre-irradiated with blue or UV-A light, and then incubated with lysates of yeast cells transfected with V5-tagged SMS2 or an empty vector (EV) (*Figure 3a,b*). A clickable ceramide analogue lacking the azobenzene moiety, **cCer** (*Figure 1c*), served as control. Next, the lipids were extracted, click-reacted with a fluorophore, separated by thin layer chromatography (TLC) and analyzed for fluorescence. To detect the clickable lipids, we initially used 3-azido-7-hydroxycoumarin as a fluorogenic click-reagent (*Sivakumar et al., 2004*; *Thiele et al., 2012*). However, this resulted in a distance-dependent intramolecular quenching of the coumarin fluorescence by the caCer azobenzene moiety (*Figure 3—figure supplement 1*). As Alexa-647 has no spectral overlap with the azobenzene, Alexa-647-azide was used in all subsequent click reactions. All four caCers and **cCer** were converted into SM when incubated with lysates of SMS2-expressing yeast cells (*Figure 3c*). No conversion was observed in lysates of control (EV) cells. Strikingly, the *cis* (UV-A-irradiated) isomers of both **caCer-1** and **caCer-2** were more efficiently metabolized by SMS2 than their corresponding *trans* (dark-adapted or blue-irradiated) isomers. **caCer-3** behaved similarly to **caCer-1** and **caCer-2**, except that its blue irradiation led to a higher metabolic conversion by SMS2 (*Figure 3c*). In contrast, the SMS2-mediated conversion of both **caCer-4** and **cCer** was independent of light treatment. The same trends were observed when the click reaction was omitted, and the azobenzene-containing lipids were visualized using the UV-absorbing properties of the azobenzene group (*Figure 3—figure supplement 2*).

To exclude the possibility that the light-dependent metabolic conversion of caCers was merely due to differences in the efficiency by which externally added *trans*- and *cis*-isoforms were incorporated into SMS2-containing membranes, we produced SMS2 cell-free in the presence of liposomes already containing the caCer probe (*Figure 3d,e*). Here, we focused on **caCer-1**, as the metabolic conversion of this compound in SMS2-containing lysates was most strongly influenced by light (*Figure 3c*). During the liposome-coupled translation of SMS2 mRNA overnight at 26°C, only trace amounts of dark-adapted **caCer-1** were converted into SM (*Figure 4f*, t = 0). Extending the incubation by 1 hr at 37°C in the dark resulted in only a minor increase in the amount of SM formed. However, extending the incubation under pulsed UV-A illumination led to a rise in the amount of SM produced. In contrast, blue illumination did not enhance SM production relative to the dark-adapted liposomes. No such light-induced fluctuations in SM production were observed with liposomes containing **cCer**. Omission of SMS2 mRNA abolished SM formation altogether (*Figure 3f*). Collectively, these results indicate that light-induced conformational changes in caCers have an acute impact on their metabolic conversion by SMS2 in both synthetic and cellular membranes.

## Optical manipulation of SM biosynthesis in living cells

We next examined whether metabolic conversion of caCers by SMS2 can be controlled in a reversible manner using the temporal precision of light. *Cis*- or *trans*-isomers of **caCer-1** and **caCer-3** were added to lysates of SMS2-expressing yeast cells, and the reactions were then incubated at 37°C under different light regimes. After each 10 min period, the caCer configuration was switched by illuminating the reactions with blue or UV-A light. The amount of SM formed was monitored over time by TLC analysis of Alexa-647-clicked reaction samples. The efficiency by which both **caCer-1** and **caCer-3** were converted into SM increased instantly upon their isomerization to *cis* with UV-A light (*Figure 4*). Conversely, isomerization back to *trans* with blue light resulted in an abrupt drop in the SM production rate. This block in SM production could be reversed again by illuminating the samples with UV-A light. These trends could be repeated over two cycles with two different light regimes (*Figure 4a,b*), thus enabling the generation of time-resolved pulses of SM synthesis. In line with our foregoing data, the light-induced changes in SM production rates were more pronounced for **caCer-1** than for **caCer-3,** and metabolic conversion of **cCer** into SM was not affected by UV-A or blue irradiation (*Figure 4a,b*).

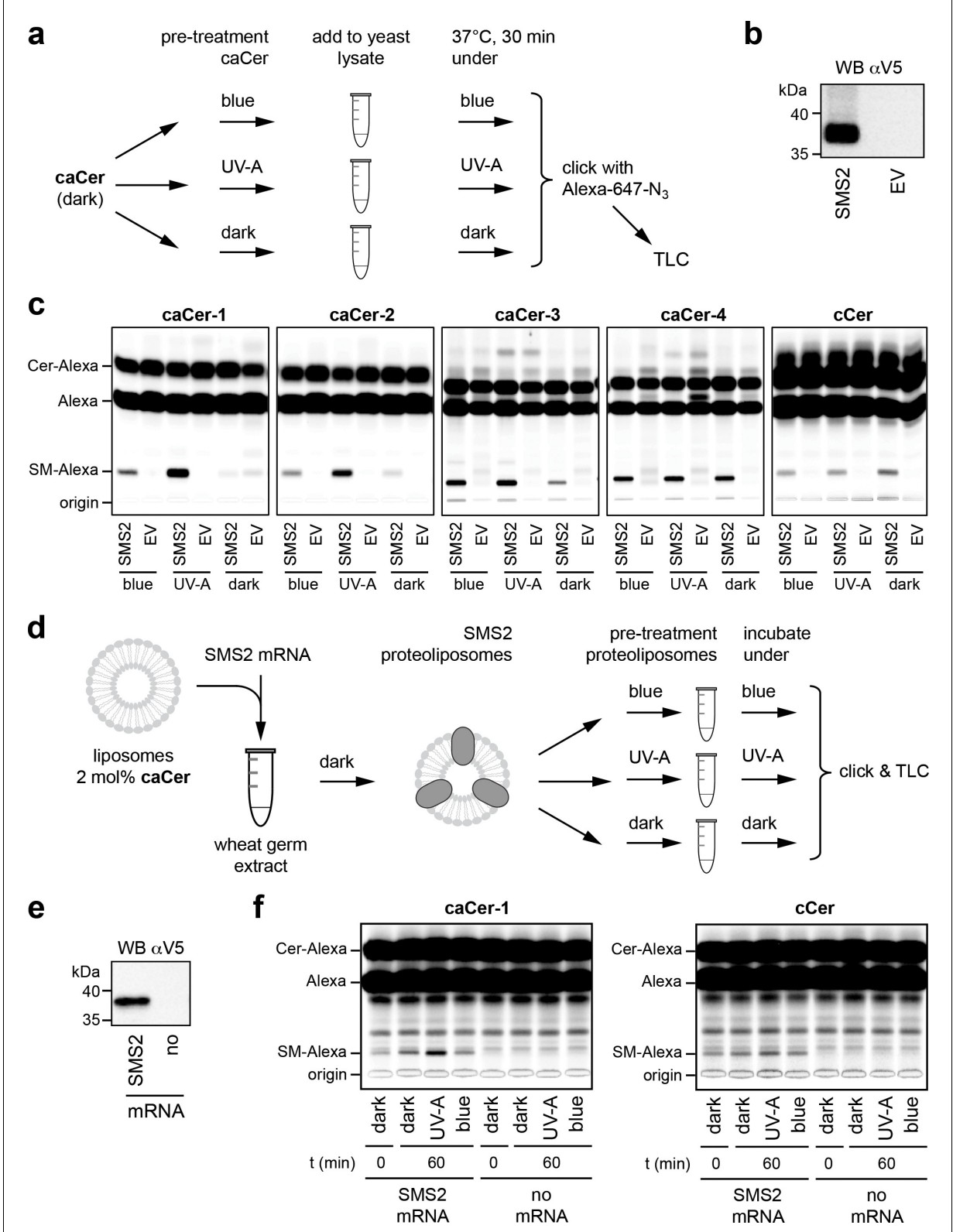

**Figure 3.** caCers are light-sensitive substrates of sphingomyelin synthase SMS2. (a) Blue, UV-A or dark-adapted **caCer**s were incubated with lysates of control or SMS2-expressing yeast cells for 30 min at 37°C and their metabolic conversion to SM was determined by TLC analysis of total lipid extracts click-reacted with Alexa-647. (b) Lysates of yeast cells transfected with empty vector (EV) or V5-tagged SMS2 were analyzed by immunoblotting with an anti-V5 antibody. (c) Lysates of control (EV) and SMS2-expressing yeast cells were incubated with **caCer**s or **cCer** as outlined in (a). Reaction samples

*Figure 3 continued on next page*

*Figure 3 continued*

were subjected to lipid extraction, click-reacted with Alexa-647 and analyzed by TLC. (**d**) SMS2 was produced cell-free in the dark at 26°C in the presence of **caCer**-containing liposomes and then incubated at 37°C in the dark or upon illumination with blue or UV-A light. Reaction samples were subjected to lipid extraction, click-reacted with Alexa-647 and analyzed by TLC. (**e**) Cell-free translation reactions with or without SMS2-V5 mRNA were analyzed by immunoblotting with an anti-V5 antibody. (**f**) SMS2 was produced cell-free in the presence of **caCer-1** or **cCer**-containing liposomes and then incubated for 0 or 60 min at 37°C as outlined in (**d**). Reaction samples were subjected to lipid extraction, click-reacted with Alexa-647 and analyzed by TLC.

DOI: https://doi.org/10.7554/eLife.43230.010

The following figure supplements are available for figure 3:

**Figure supplement 1.** Monitoring SMS2-mediated metabolic conversion of caCers and cCer using 3-azido-7-hydroxycoumarin as click reagent.
DOI: https://doi.org/10.7554/eLife.43230.011

**Figure supplement 2.** Monitoring SMS2-mediated metabolic conversion of caCers independently of fluorescent click-reagents.
DOI: https://doi.org/10.7554/eLife.43230.012

To investigate whether caCers also enable optical control over SM biosynthesis in living cells, we used SMS2-overexpressing HeLa cells generated by stable transfection with a V5-tagged SMS2 expression construct (*Figure 5a*). As expected, immunofluorescence microscopy revealed that the bulk of SMS2-V5 resides at the plasma membrane (*Figure 5b,c*) (*Huitema et al., 2004*; *Tafesse et al., 2007*). Next, control and SMS2-overexpressing HeLa cells were incubated with externally-added *cis*- (UV-A-irradiated) or *trans*- isomers (blue-irradiated or dark-adapted) of **caCer-1** and **caCer-3** for 1 hr at 37°C, and their conversion into SM was determined by TLC analysis of Alexa-647-clicked total lipid extracts. The overall yield of SM produced in SMS2-overexpressing cells was at least threefold higher than that in control cells, regardless of the caCer isomer used (*Figure 5d–g*). However, in both control and SMS2-overexpressing cells, *cis*-**caCer-1** was more efficiently metabolized into SM than the *trans*-isomer (*Figure 5d,e*), and similar results were obtained with **caCer-3** (*Figure 5f,g*).

To address whether the rate of SM production in living cells could be controlled reversibly, SMS2-overexpressing cells were incubated with a *trans* (dark-adapted) isomer of **caCer-1** or with **cCer** for 15 min at 37°C in the dark. Cells were then flash-illuminated with blue followed by UV-A light or vice-versa, and then incubated for another 20 min in the dark. Cells that did not receive any light treatment and that were kept in the dark throughout the incubation period served as baseline for the conversion of the *trans*-isomer only. The amount of SM formed was then determined by TLC analysis of Alexa-647-clicked total lipid extracts. Cells treated first with UV-A followed by blue-light produced similar amounts of SM as cells kept in the dark (*Figure 5h,i*). However, treating cells first with blue light followed by UV-A light caused a marked (3–5-fold) increase in SM production. In contrast, production of SM from cCer was not affected by light treatment, demonstrating that UV-A irradiation itself did not affect SM-production in HeLa cells. Taken together, these results demonstrate the suitability of caCers as photoswitchable substrates for manipulating SM biosynthesis in living cells with the temporal precision of light.

## caCers enable optical manipulation of GlcCer biosynthesis

Finally, we addressed whether caCers can be used as light-sensitive substrates of other sphingolipid biosynthetic enzymes. We employed a yeast strain expressing human glucosylceramide synthase (GCS; *Figure 6a*), a Golgi-resident enzyme structurally unrelated to SMS2 that generates glucosylceramide (GlcCer) by transferring glucose from UDP-glucose to ceramide. When incubated with lysates of GCS-expressing yeast cells in the presence of UDP-glucose, UV-A-irradiated **caCer-1** was converted to GlcCer (*Figure 6b*). No GlcCer was formed when UDP-glucose was omitted or when yeast lysates that lack GCS were used. Analogous to our results with SMS2, metabolic conversion of **caCer-1** by GCS was light sensitive, with the *cis*-isomers being more readily converted to GlcCer than the *trans*-isomers (*Figure 6c,d*). To address whether the rate of GlcCer production from caCers could be reversibly manipulated, GCS-containing yeast lysates were incubated with a *trans* (dark-adapted) isomer of **caCer-1** or with **cCer** for 15 min at 37°C in the dark. The lysates were then flash-illuminated with blue light followed by UV-A light or vice-versa, and then incubated for another 20 min in the dark. Subjecting *trans*-**caCer-1**-containing lysates to an 'off-ON' switching regime (blue then UV-A) enabled GlcCer production (*Figure 6e*). In contrast, keeping the lysates in the dark or

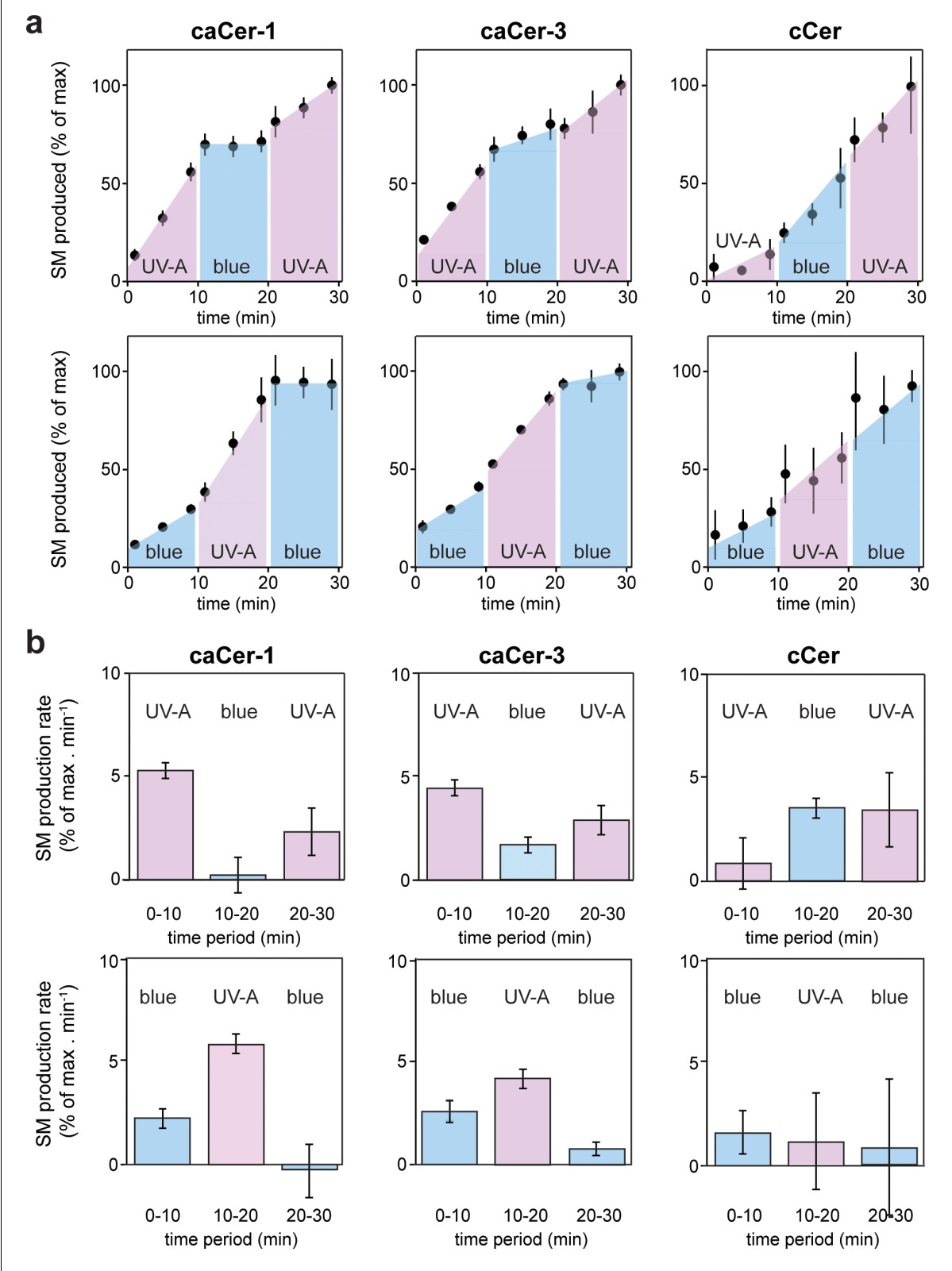

**Figure 4.** Metabolic conversion of caCers by SMS2 in cell lysates can be switched on and off by light. (a) **caCer-1**, **caCer-3** or **cCer** were incubated with lysates of SMS2-expressing yeast cells at 37°C under UV-A or blue illumination. After each 10 min period, the **caCer** configuration was switched by illuminating the reactions with blue or UV-A light. Reaction samples were taken at the indicated time points, subjected to lipid extraction, click-reacted with Alexa-647 and then analyzed by TLC. Presented are the relative amounts of SM formed per time point. (b) Presented are the relative SM
*Figure 4 continued on next page*

*Figure 4 continued*

production rates (a.u. of SM formed per min) calculated from the time points presented in (a). Data shown are average values ± s.d. from four technical replicates (*n* = 4).

DOI: https://doi.org/10.7554/eLife.43230.013

The following source data is available for figure 4:

**Source data 1.** Quantitation of metabolic conversion of caCers by SMS2 in cell lysates.

DOI: https://doi.org/10.7554/eLife.43230.014

subjecting them to an 'on-OFF' switching regime (UV-A then blue) blocked GlcCer production. In lysates containing **cCer**, GlcCer production was independent of light (*Figure 6c–e*). To conclude, we produced GCS cell-free in the presence of UDP-glucose and liposomes that contained either dark-adapted **caCer-1** or **cCer** (*Figure 6f*). While only small amounts of GlcCer were formed when **caCer-1**-containing proteoliposomes were kept in the dark, GlcCer production was strongly stimulated by UV-A illumination (*Figure 6g*). In contrast, GlcCer production in **cCer**-containing proteoliposomes was insensitive to UV-A illumination. Importantly, these results demonstrate that the application of caCers as light-sensitive substrates is not restricted to one particular class of ceramide metabolic enzymes.

## Discussion

Studies of lipid function traditionally revolve around an extended central dogma of molecular biology: that proteins control cellular lipid pools by catalyzing lipid metabolism and transport. As such, interrogating lipid function by manipulating gene expression or protein activity is intrinsically slow and indirect. Photo-activated lipids allow one to probe lipid function in a more acute and direct manner using the unmatched temporal precision of light. Here, we report on caCers, a new generation of clickable and azobenzene-containing ceramide analogs that enable optical control over sphingolipid production in living cells. Isomerization of the azobenzene photoswitch triggers conformational changes in the carbon chains of caCers that either block or facilitate their metabolic conversion by SMS2 and GCS, two key sphingolipid biosynthetic enzymes in mammals. Our finding that these light-induced effects are instant, reversible and occur in native cellular membranes opens up unprecedented opportunities to manipulate the metabolic fate and biological activity of sphingolipids at the subcellular level in real time.

Using synthetic and cellular membrane systems as experimental models, we found that *cis*-isomers of **caCer-1**, **caCer-2** and **caCer-3** are more readily metabolized by SMS2 than their corresponding *trans*-isomers. These observations may seem counterintuitive at first, as *trans*-caCers resemble natural ceramides more closely than *cis*-caCers, possessing kinked carbon chains instead of straight ones. One explanation for this may be that the *trans*-azobenzene moiety interferes with high-affinity binding of caCers to the ceramide binding pocket in SMS2, for instance because the enzyme preferentially accommodates substrates with bent carbon chains (*Figure 7a*). However, this theory conflicts with the finding that SM synthases are relatively tolerant toward structural deviations in the ceramide backbone and readily metabolize ceramide analogous with distinct functional groups in the sphingoid base or *N*-acyl chain (*Koval and Pagano, 1991*; *Huitema et al., 2004*; *Haberkant et al., 2016*). Moreover, the unrelated enzyme GCS also converts *cis*-caCers more rapidly, suggesting that a general biophysical property of the caCers underlies this phenomenon.

Based on our findings in SLBs, we hypothesize that the impact of *cis-trans* isomerization on the metabolic fate of caCers may be caused by light-induced alterations in their lateral packing (*Figure 7b*). According to this model, *trans*-caCers form tightly packed clusters due to favorable intermolecular stacking interactions among the flat *trans*-azobenzene groups, thereby reducing their availability for enzymatic conversion. On isomerization to *cis*, the kinked caCer molecules become dispersed throughout the membrane, making them more accessible for enzymatic conversion. This idea is supported by our finding that *trans*-caCers preferentially localize in liquid-ordered ($L_o$) domains in model bilayers (*Figure 2*) (*Frank et al., 2016b*). **caCer-4** forms an exception, as its isomerization to *trans* promotes association with $L_o$ domains without having an obvious impact on SM biosynthesis. While both **caCer-3** and **caCer-4** have azobenzene-containing sphingoid bases,

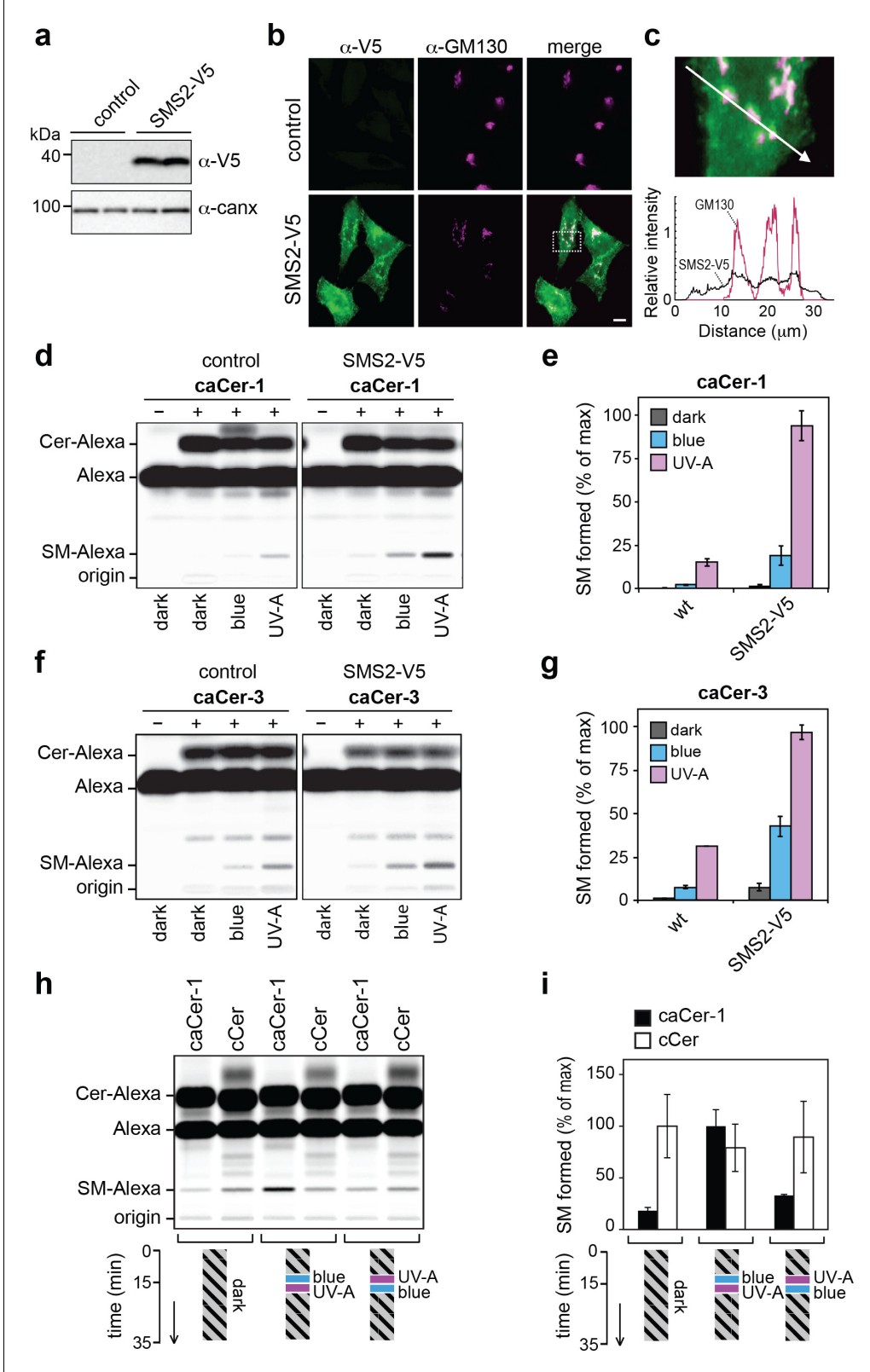

**Figure 5.** caCers enable optical control of SMS2-mediated SM biosynthesis in living cells. (a) HeLa cells stably transfected with V5-tagged SMS2 were analyzed by immunoblotting using anti-V5 and anti-calnexin antibodies. Untransfected HeLa cells served as control. (b) Control and SMS2-V5-expressing HeLa cells were fixed, co-stained with antibodies against the V5 epitope and Golgi marker GM130, and then visualized by fluorescence microscopy. Scale bar, 10 μm. (c) Intensity plots along the path marked by the white arrow, showing overlap between anti-V5 (green) and anti-GM130

*Figure 5 continued on next page*

*Figure 5 continued*

(magenta) channels. (**d**) Blue, UV-A or dark-adapted **caCer-1** was incubated with control or SMS2-V5-expressing HeLa cells for 1 hr at 37°C. Metabolic conversion of **caCer-1** to SM was determined by TLC analysis of total lipid extracts click-reacted with Alexa-647. (**e**) Quantitative analysis of SM formed from **caCer-1** by cells treated as in (**c**). (**f**) Blue, UV-A or dark-adapted **caCer-3** was incubated with control or SMS2-V5-expressing HeLa cells for 1 hr at 37°C and its metabolic conversion to SM was determined as in (**c**). (**g**) Quantitative analysis of SM formed from **caCer-3** by cells treated as in (**e**). (**h**) **caCer-1** or **cCer** were incubated with SMS2-V5-expressing HeLa cells at 37°C in the dark. After 15 min, cells were flash-illuminated by blue light followed by UV-A or vice versa and then incubated for another 20 min. Cells kept in the dark for the entire incubation period served as control. Metabolic conversion of **caCer-1** or **cCer** to SM was determined by TLC analysis of total lipid extracts click-reacted with Alexa-647. (**i**) Quantitative analysis of SM formed from **caCer-1** or **cCer** by cells treated as in (**g**). Data shown are mean values ± s.d. from three biological replicates (*n* = 3).
DOI: https://doi.org/10.7554/eLife.43230.015

The following source data is available for figure 5:

**Source data 1.** Quantitation of metabolic conversion of pretreated caCer1 by SMS2 in living HeLa cells.
DOI: https://doi.org/10.7554/eLife.43230.016
**Source data 2.** Quantitation of metabolic conversion of pretreated caCer3 by SMS2 in living HeLa cells.
DOI: https://doi.org/10.7554/eLife.43230.017
**Source data 3.** Quantitation of live manipulation of metabolic conversion of caCer1 and cCer by SMS2 in living HeLa cells.
DOI: https://doi.org/10.7554/eLife.43230.018

---

**caCer-4** is unique among all caCers in that its azobenzene photoswitch is positioned close to the end of the carbon chain. Conceivably, isomerization of this compound to *trans* may lead to a less pronounced clustering in cellular membranes.

A distinct advantage of caCers over caged ceramide analogs (*Kim et al., 2016*; *Höglinger et al., 2014*; *Höglinger et al., 2015*) is that their activity can be readily switched off by light. This property enabled us to generate time-resolved pulses of SM synthesis at the plasma membrane of living cells. In comparison to traditional pulse-chase approaches, our method offers superior temporal resolution for manipulating sphingolipid levels, and could be combined with patterned illumination to generate controlled sphingolipid pulses in subpopulations of cells or in subcellular organelles. An attractive prospect is to use *cis-trans* isomerization to bring caCers within reach of the binding pocket of a ceramide signaling protein, or to pull them away, thus enabling instant and reversible control over ceramide-operated signaling pathways (*Figure 7*). Accordingly, caCers may find use in dissecting the causal roles of ceramides in apoptosis, as tumor suppressors, and as antagonists of insulin signaling (*Hannun and Obeid, 2018*; *Jain et al., 2017*; *Park et al., 2016*; *Meikle and Summers, 2017*). In fact, light-induced changes in lateral packing may be a previously unnoted characteristic of other azobenzene-containing signaling lipids like DAGs, where the *cis*-isomers correspond to the bioactive form (*Frank et al., 2016a*; *Leinders-Zufall et al., 2018*).

Application of caCers in living cells is hampered by their relatively poor aqueous solubility. However, the development of photoswitchable sphingoid bases may provide a way to circumvent limitations associated with the relative inefficient uptake of caCers by cells, thus expanding opportunities for manipulating the metabolic fate and signaling activity of sphingolipids by light. Ceramides are the precursors of a large collection of complex sphingolipids with critical roles in sustaining mechanical stability, molecular sorting, cell signaling, migration and adhesion. Consequently, caCers provide attractive scaffolds for creating a new toolbox of photoswitchable sphingolipids to enable optical control over a wide array of sphingolipid-mediated cellular processes.

## Materials and methods

### General chemical synthesis

Clickable ceramide (**cCer**) was synthesized as previously described (*Kol et al., 2017*). Unless otherwise stated, all reactions were performed with magnetic stirring under a positive pressure of nitrogen or argon gas. Tetrahydrofuran (THF) and diethyl ether (Et$_2$O) were distilled over sodium benzophenone under nitrogen atmosphere prior to use. Dichloromethane (CH$_2$Cl$_2$) and triethylamine (Et$_3$N) were distilled over calcium hydride under a nitrogen atmosphere. *N,N*-dimethylformamide (DMF), toluene (PhMe), dioxane and methanol (MeOH) were purchased from Acros Organics as 'extra dry' reagents under inert gas atmosphere and stored over molecular sieves. Ethyl acetate

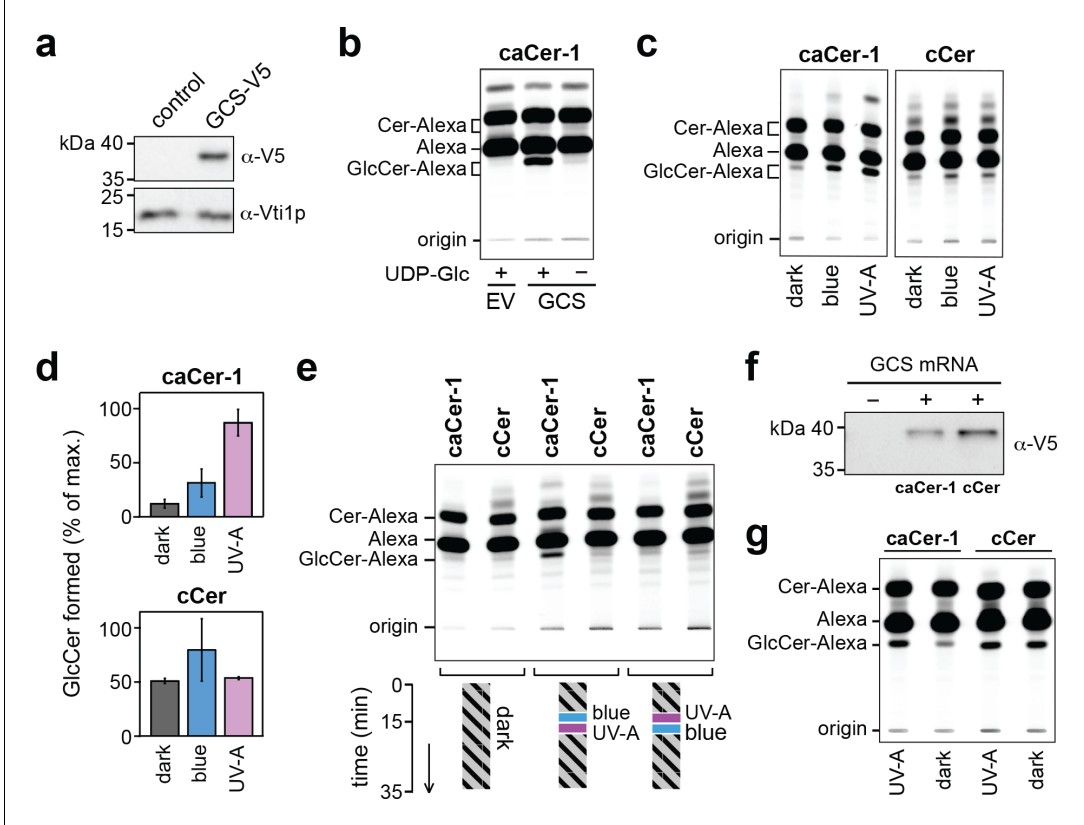

**Figure 6.** caCer-1 is a light-sensitive substrate of glucosylceramide synthase GCS. (**a**) Lysates of yeast cells transfected with V5-tagged GCS or empty vector (EV; control) were analyzed by immunoblotting using anti-V5 and anti-Vti1p antibodies. (**b**) UV-A pretreated **caCer-1** (25 µM) was incubated with lysates of control or GCS-expressing yeast cells for 30 min at 37°C in the presence or absence of 1 mM UDP-glucose. Reactions were subjected to lipid extraction, click-reacted with Alexa-647 and then analyzed by TLC. (**c**) UV-A, blue or dark pretreated **caCer-1** and **cCer** (25 µM, each) were incubated in the dark for 30 min at 37°C with lysates of GCS-expressing yeast in the presence of 1 mM UDP-glucose and analyzed as in (**b**). (**d**) Quantitative analysis of GlcCer formed from **caCer-1** or **cCer** in lysates treated as in (**c**). (**e**) Dark-adapted **caCer-1** or **cCer** (25 µM, each) and UDP-glucose (1 mM) were incubated with lysates of GCS-expressing yeast in the dark at 37°C. After 15 min, lysates were flash-illuminated by blue light followed by UV-A or vice versa and then incubated for another 20 min. Lysates kept in the dark for the entire incubation period served as control. GlcCer formation was determined by TLC analysis of total lipid extracts click-reacted with Alexa-647. (**f**) Cell-free translation reactions with or without GCS-V5 mRNA were carried out overnight in the dark at 26°C. GCS was expressed in the presence of liposomes containing **caCer-1** or **cCer** and then analyzed by immunoblotting using an anti-V5 antibody. (**g**) GCS produced cell-free as in (**f**) was flash-illuminated with UV-A or kept in the dark and then incubated with 1 mM UDP-glucose for 30 min at 37°C in the dark. GlcCer formation was monitored by TLC as in (**e**). Data shown are average values ± s.d. (**cCer**, $n = 2$; **caCer-1**, $n = 3$).

DOI: https://doi.org/10.7554/eLife.43230.019

The following source data is available for figure 6:

**Source data 1.** Quantitation of metabolic conversion of pretreated caCer-1 and cCer by GCS in cell lysates.
DOI: https://doi.org/10.7554/eLife.43230.020

(EtOAc), pentane, Et$_2$O, CH$_2$Cl$_2$ and MeOH used specifically for extraction and flash column chromatography were purchased at technical grade from commercial sources and distilled under reduced pressure. Solvents and reagents were used as received from commercial sources (Sigma-Aldrich, Tokyo Chemical Industry Co., Alfa Aesar, Acros Organics, Strem Chemicals). Reactions were monitored by thin-layer chromatography (TLC) using silica gel F254 pre-coated glass plates (*Merck*) and visualized by exposure to ultraviolet light ($\lambda$ = 254 nm) or by staining with aqueous potassium permanganate (KMnO$_4$) solution (7.5 g KMnO$_4$, 50 g K$_2$CO$_3$, 6.25 mL aqueous 10% NaOH, 1000 mL distilled H$_2$O), aqueous acidic ceric ammonium molybdate (IV) (CAM) solution (2.0 g Ce(NH$_4$)$_4$(SO$_4$)$_4$·2H$_2$O, 48 g (NH$_4$)$_6$Mo$_7$O$_{24}$·4H$_2$O, 60 mL concentrated sulfuric acid, 940 mL distilled H$_2$O) or a butanolic ninhydrin solution (13.5 g ninhydrin, 900 mL *n*-BuOH, 27 mL acetic acid) followed

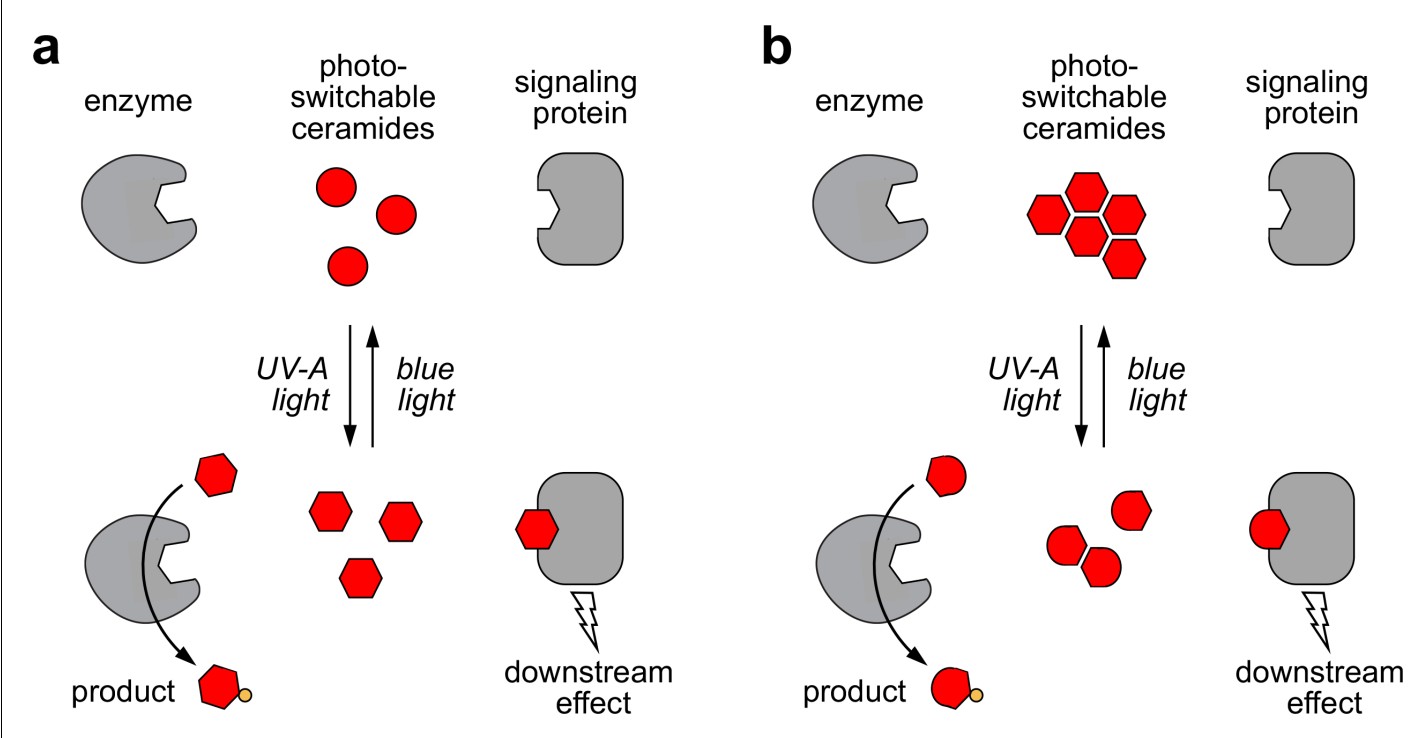

**Figure 7.** Models of how photoswitchable ceramides may enable optical control over sphingolipid biosynthesis and signaling. (**a**) UV-A or blue light trigger alterations in the curvature of the carbon chains of azobenzene-containing ceramides (caCers), which reversibly enhance or reduce their affinity for ceramide metabolic enzymes and signaling proteins. (**b**) Blue light triggers self-assembly of photoswitchable ceramides (caCers) into tightly packed clusters driven by intermolecular stacking of the flat *trans*-azobenzene groups. In contrast, UV-A light causes caCers to disperse in the plane of the membrane as the bent *cis*-azobenzene groups disrupt intermolecular stacking. These light-induced changes in lateral packing of caCers reversibly enhance or reduce their availability for metabolic conversion or stimulating ceramide signaling proteins.

DOI: https://doi.org/10.7554/eLife.43230.021

by heating with a heat gun (150–600°C). Flash column chromatography was performed using silica gel (60 Å, 40–63 µm, *Merck*).

Proton ($^1$H) and carbon ($^{13}$C) nuclear magnetic resonance spectra were recorded on a *Bruker* Avance III HD 400 MHz spectrometer equipped with a CryoProbe or a *Varian* VXR400 S spectrometer. Proton chemical shifts are expressed in parts per million (ppm, δ scale) and referenced to residual undeuterated solvent signals. Carbon chemical shifts are expressed in parts per million (ppm, δ scale) and referenced to the central carbon resonance of the solvent. The reported data is represented as follows: chemical shift in parts per million (ppm, δ scale) (multiplicity, coupling constants *J* in Hz, integration intensity, proton assignment). Abbreviations used for analysis of multiplets are as follows: s (singlet), br s (broad singlet), d (doublet), t (triplet), q (quartet), p (pentet), h (hextet), and m (multiplet). Variable temperature NMR spectroscopy was performed at the *Ludwig-Maximilians-Universität* NMR facility.

IR spectra were recorded on a PerkinElmer Spectrum BXII FTIR spectrometer equipped with an attenuated total reflection (ATR) measuring unit. IR data is recorded in frequency of absorption (wavenumber in cm$^{-1}$). Mass spectrometry (MS) experiments were performed at high resolution on a Thermo Finnigan MAT 95 (electron ionization [EI] double-focusing magnetic sector mass spectrometer) or on a Thermo Finnigan LTQ FT (electrospray ionization [ESI] linear ion trap-based Fourier transform ion cyclotron resonance mass spectrometer) instrument at the *Ludwig-Maximilians-Universität* mass spectrometry facility. Melting points were measured using a *Stanford Research Systems* MPA120 Automated Melting Point Apparatus in open capillaries and are uncorrected.

## Supported lipid bilayer formation

N-Stearoyl-D-erythro-sphingosine (C18:0-Cer), N-stearoyl-D-erythro-sphingosylphosphoryl-choline (C18:0-SM), 1,2-dioleoyl-sn-glycero-3-phosphocholine (DOPC) and cholesterol were purchased from Avanti Polar Lipids (Alabaster, AL, USA). We prepared supported lipid bilayers (SLBs) by deposition and fusion of small unilamellar vesicles (SUVs) as described elsewhere (Chiantia et al., 2006). SUVs composed of DOPC:cholesterol:SM:caCer, containing additional 0.1 mol% ATTO655-DOPE (ATTO Technology GmbH, Siegen, Germany), were obtained by bath sonication of multilamellar vesicles. SUV suspensions (1 mM total lipid concentration in buffer containing 10 mM HEPES, 150 mM NaCl, pH 7.4) were deposited in the presence of 2 mM CaCl$_2$ on freshly-cleaved mica glued to glass cover-slips. The samples were incubated at 65 °C for 30 min, rinsed with buffer and then allowed to cool slowly to room temperature for at least 1 hr.

## Combined atomic force and confocal microscopy

Combined atomic force and confocal microscopy was performed on a JPK Instruments Nanowizard III BioAFM and Nanowizard Ultra (Berlin, Germany) mounted on a Zeiss LSM510 Meta laser scanning confocal microscope (Jena, Germany). High-speed AFM in AC mode was done with the Nanowizard Ultra head, utilizing USC-F0.3-k0.3 ultra-short cantilevers from Nanoworld (Neuchâtel, Switzerland) with typical stiffness of 0.3 N/m. The cantilever oscillation was tuned to a frequency of 100–150 kHz and the amplitude kept below 10 nm. The scan rate was set to 25–150 Hz. Images were acquired at 256 × 256 pixel resolution. All measurements were performed at room temperature. The force applied on the sample was minimized by continuously adjusting the set point and gain during imaging. Height, error, deflection and phase-shift signals were recorded and images were line-fitted as required. Data was analyzed using JPK data processing software Version 5.1.4 (JPK Instruments) and Gwyddion Version 2.30 (Czech Metrology Institute). For the confocal measurements, a 633 nm He-Ne laser (to excite the 0.1 mol% ATTO655-DOPE added to the lipid mixtures) and a 40 × NA 1.2 UV-VIS-IR C Apochromat water-immersion objective were used. All measurements were performed at room temperature. Images were typically acquired with a 512 × 512 pixel resolution at a scan rate of 3.2 µs/pixel and using a 1 Airy pinhole. Images were further processed with Fiji software (http://fiji.sc/Fiji).

## Compound switching on supported lipid bilayers

Compound switching for combined atomic force and confocal microscopy was achieved using a CoolLED pE-2 LED light source (Andover, United Kingdom) for illumination at 365 and 470 nm. The light source was operated at a maximum of 80% power. The light beam was guided by a fiber-optic cable directly through the objective of the microscope via a collimator at the backport side of the microscope.

## Preparation of yeast lysates

The open reading frames of human SMS2 (uniProt entry Q8NHU3) and GCS (uniProt entry Q16739) were PCR amplified and cloned into the pYES2.1/V5-His TOPO vector (Invitrogen) according to the manufacturer's instructions. Yeast strain IAY11 (MATα, ade2-1 trp1-1 can1-100 leu2-3,112 his3-11,15 ura3-52 ade3-Δ853) was transformed with pYES2.1/SMS2-V5-His, pYES2.1/GCS-V5-His, or an empty vector control. For SMS2 expression, yeast was grown at 30°C in synthetic medium containing 2% (w/v) galactose to early mid-logarithmic phase. For GCS expression, yeast was grown in synthetic medium with 2% (w/v) glucose to mid-log phase, washed twice with sterile water, and then grown in synthetic medium with 2% (w/v) galactose for 4 hr. Cells were collected by centrifugation and washed in ice-cold Buffer R (15 mM KCl, 5 mM NaCl, 20 mM HEPES/KOH pH 7.2). The wet cell pellet (2 g) was resuspended in buffer R (5 mL) containing protease inhibitor cocktail (PIC, 1 µg/mL aprotinin, 1 µg/mL leupeptin, 1 µg/mL pepstatin, 5 µg/mL antipain, 1 mM benzamidine and 1 mM phenylmethanesulfonyl fluoride). The cells were lysed by vigorous vortexing with glass beads (3 g) at 4 °C with intermittent cooling on ice. Post-nuclear supernatants were prepared by centrifugation at 700 x$g$ for 10 min at 4 °C. The PNS of GCS-expressing yeast was centrifuged at 100,000 x$g$ for 1 hr at 4°C and the resulting membrane pellet was resuspended in buffer R. After addition of 0.11 vol of glycerol, the PNS and resuspended membranes were aliquoted, snap-frozen in liquid nitrogen and

stored at −80 °C. Protein concentration of PNS and membrane samples was determined by Bio-Rad Protein Assay (Bio-Rad GmbH, Munich, Germany).

## Enzyme activity assay on yeast lysates

PNS of control or SMS2-expressing yeast were diluted to 0.3 mg/mL (*Figure 3c*) or 0.6 mg/mL total protein (*Figure 4*) in Buffer R supplemented with PIC and 0.5 mM N-ethylmaleimide, and kept on ice in 4 mL brown glass vials. Stocks of cCer and caCers (2 mM in EtOH) were stored in the dark at −20 °C in 1.8 mL brown glass vials with ethylene-tetrafluoroethylene (ETFE)-coated rubber seals. After pre-treatment with high intensity illumination at 365 or 470 nm using a CoolLED pE-2 (80% power, 1 min, on ice), cCer and caCers were added to 400 µl PNS at a final concentration of 25 µM and immediately vortexed. After 10 min on ice in the dark, the reactions were shifted to 37 °C for 30 min and kept in the dark or subjected to low intensity flash illumination (pulse of 75 ms with 15 s interval) with UV-A (~365 nm) or blue light (~460 nm) from the top of the reaction tube using a home-built, 24 well plate-compatible pulsed LED illumination setup (*Borowiak et al., 2015*) while shaking at 30 rpm. For photoswitch experiments (*Figure 4*, time point t = 10 min and t = 20 min), the reactions were subjected to high intensity illumination at 365 or 470 nm with a CoolLED pE-2 (80% power) for 20 s at 25 °C and then incubated again at 37 °C in the dark. Membrane suspensions of control or GCS-expressing yeast were diluted to 0.6 mg/mL total protein in Buffer R and supplemented with 1 mM UDP-glucose. Light-treated or dark-adapted cCer and caCers were added to 400 µl of membrane suspension at a final concentration of 25 µM in 4 mL brown glass vials and then incubated at 37 °C for 30 min in the dark with gentle shaking. Reactions were stopped by addition of 3.75 vol CHCl$_3$:MeOH (1:2 vol:vol) and stored at −20 °C.

## Enzyme activity assays on cell-free-produced SMS2 and GCS

The open reading frames of human SMS2 and GCS were PCR amplified in-frame with a *C*-terminal V5 epitope, cloned into wheat germ expression vector pEU-Flexi, and subjected to in vitro transcription and translation as described (*Kol et al., 2017*). Translation of SMS2 and GCS transcripts was incubated overnight at 26 °C in the dark in the presence of liposomes (2 mM) comprising egg PC: egg PE:wheatgerm PI:cCer/caCer (40:40:20:2 molar ratio). Next, translation reactions were split into four aliquots. One aliquot served as the t = 0 control, and the other three were treated either with UV-A or blue light, or kept in the dark, and subsequently incubated for 1 hr at 37 °C under pulsed LED illumination or in the dark to record SMS2 activity. For GCS activity, 1 mM UDP-glucose was added directly prior to splitting and light treatment, and the reactions were incubated for 30 min at 37 °C. Reactions were stopped by addition of 3.75 vol of CHCl$_3$:MeOH (1:2 vol:vol) and stored at −20 °C.

## Enzyme assay on HeLa cells

HeLa cells (ATTC CCL-2) were cultured in DMEM supplemented with 9% FBS (PAN Biotech P40-47500) at 37 °C under 5% CO$_2$. HeLa cells were mycoplasma free. HeLa cells stably transfected with pcDNA3.1-SMS2-V5-His-TOPO (*Huitema et al., 2004*) were cultured in the presence of 400 µg/mL G418 (Biochrom A291-25). Cells were seeded at 200,000–300,000 cells per well of a 6-well plate (Greiner CellStar 657160). After 8 hr, 4 mM sodium butyrate (SantaCruz SC 202341A, 1 M stock in H$_2$O) was added. After another 16 hr, the cells were washed with PBS and then incubated in 1 mL phenol red-free Optimem (Gibco 11058–021, Thermo Scientific) per well. Stock solutions of cCer and caCers (2 mM in DMSO, Applichem A3672) were pre-treated with high intensity illumination at either 365 or 470 nm with a CoolLED pE-2 (80% power) for 1 min on ice or kept in the dark, and then immediately added to the Optimem on the cells under reduced light conditions. Incubations were performed at 37 °C under 5% CO$_2$ in the dark (*Figure 5c–f*). Cells were trypsinized, re-suspended and harvested in 1 mL PBS supplemented with PIC. Cells were then pelleted at 900 x*g* for 5 min at 4 °C. After removal of 900 µl of supernatant, the cell pellet was re-suspended in the remaining 100 µl and 375 µl CHCl$_3$:MeOH (1:2 vol:vol) was added before samples were stored at −20 °C.

For live switch experiments (*Figure 5g,h*), cells were grown and treated with butyrate in T75 flasks, as described above. Cells were trypsinized, collected in DMEM, pelleted at 600 x*g* for 5 min at RT, washed with phenol red-free Optimem and then resuspended at ±8×10$^6$ cells/mL in phenol red-free Optimem. From this cell suspension, 100 µl aliquots were transferred to 1.8 mL brown vials,

supplemented with 60 µM dark-adapted cCer or caCer-1 (from 2 mM stocks in DMSO) and incubated at 37 °C under 5% $CO_2$ in the dark with gentle shaking at 600 rpm. Light treatment was performed by high intensity illumination for 30 s at 20 °C with a CoolLED pE-2 at either 365 or 470 nm, 80% power output. Reactions were stopped by addition of 375 µl $CHCl_3$:MeOH (1:2 vol:vol) and stored at −20 °C.

## Immunoblot analysis

Expression of V5-tagged SMS2 or GCS was verified by immunoblot analysis using a mouse monoclonal anti-V5 antibody (ThermoFisher, cat. no. R960-25, 1:4000). Rabbit polyclonal anti-calnexin (Santa Cruz, cat. no. SC11397, 1:2000) and anti-Vti1p antibodies (kind gift from Christian Ungermann, University of Osnabrück, 1:2000) served as controls. Goat-anti-mouse and goat-anti-rabbit IgG HRP conjugates were obtained from ThermoFisher (cat. no. 31430, 1:4000) and Biorad (cat. no. 1706515, 1:4000), respectively. For detection, Pierce ECL reagent (ThermoFisher, cat. no. 32106) was used. Immunoblots were processed using a ChemiDoc XRS + system (Bio-Rad) and ImageLab software.

## Lipid extraction

Lipid extractions were performed in Eppendorf Protein LoBind tubes with a reference volume (one vol) of 100 µl (sample) in 3.75 vol (375 µl) $CHCl_3$:MeOH. After centrifugation at 21,000 x*g* for 10 min at 4 °C, the supernatant was collected and transferred to a fresh tube containing one vol $CHCl_3$ and 1.25 vol 0.45% NaCl to induce phase separation. After vigorous vortexing for 5 min at RT and subsequent centrifugation (5 min, 21,000 x*g*, RT), the organic phase was transferred to a fresh tube containing 3.5 vol MeOH:0.45% NaCl (1:1 vol:vol). After vigorous vortexing for 5 min at RT and subsequent centrifugation (5 min, 21,000 x*g*, RT), the organic phase (1.8 vol) was collected and used for derivatization with clickable fluorophores.

## Click reactions

Clickable fluorophores 3-azido-7-hydroxycoumarin (Jena Bioscience) and Alexa-647-azide (Thermo Fisher) were dissolved in $CH_3CN$ to a final concentration of 10 mM and 2 mM, respectively, and stored at −20 °C. Lipid extracts were transferred to Eppendorf Protein LoBind tubes, dried down in a speedvac and dissolved in 10 µl $CHCl_3$ (initial experiments) or directly dissolved in click mix (later experiments), as indicated. The alkyne-functionalized lipids in 10 µl $CHCl_3$ were clicked to 3-azido-7-hydroxycoumarin or Alexa647-azide by incubation with 43 µl of a freshly prepared click reaction solution containing 0.45 mM of the clickable fluorophore and 1.4 mM tetrakis(acetonitrile)copper(I) tetrafluoroborate in $CH_3CN$:EtOH (3:7, vol:vol) for 4 hr at 40 °C, followed by 12 hr at 12 °C without shaking (Gaebler et al., 2013). The reaction was quenched by the addition of 150 µl MeOH, dried down in a Christ RVC 2–18 speedvac under reduced pressure from a Vacuubrand MZ 2C diaphragm vacuum pump, and dissolved in $CHCl_3$:MeOH (2:1, vol:vol). In later experiments, dried lipid films were directly dissolved in 25 µl of click mix. The click-mix was composed of $CHCl_3$:$CH_3CN$:EtOH (19:16:66, vol:vol:vol), in which Alexa-647 was present at an equimolar ratio with respect to the calculated amount of alkyne lipid (assuming 100% lipid recovery from the extraction). The tetrakis(acetonitrile)copper(I) tetrafluoroborate was present as at 20-fold molar excess with respect to the fluorophore. After incubation for 4 hr at 40 °C and 12 hr at 12 °C without shaking, the reaction mixtures were directly loaded on TLC.

## TLC analysis

Click-reacted lipid extracts were applied at 120 nl/s to NANO-ADAMANT HP-TLC plates (Macherey and Nagel) using a CAMAG Linomat 5 TLC sampler. The TLC was developed in $CHCl_3$:MeOH:$H_2O$: AcOH (65:25:4:1, vol:vol:vol:vol) using the CAMAG ADC2 automatic TLC developer operated as follows: 30 s plate pre-drying, 5 min humidity control (against saturated KSCN solution), 10 min tank saturation with eluent, after which the TLC was developed until the front reached 85 mm height, and then dried for 5 min. Coumarin-derivatized and unclicked azobenzene lipids were analyzed using a ChemiDoc XRS + imager with UV-trans-illumination (detection settings for Ethidium Bromide - standard filter) and Quantity One software (Bio-Rad). Alexa-647 derivatized lipids were visualized on TLC using a Typhoon FLA 9500 Biomolecular Imager (GE Healthcare Life Sciences) operated with 650 nm excitation laser, LPR filter, 50 µm pixel size and PMT voltage setting of 290 V. The amount of Alexa-

647-derivatized SM or GlcCer was quantified using the ImageQuant TL toolbox software (GE health-care) as a background-corrected band intensity normalized to the total clicked-lipid signal intensity (SM +Cer or GlcCer +Cer) for each lane, to correct for occasional sample-to-sample differences in click labeling efficiency. Next, the highest sample-normalized intensity of a particular time series was normalized to the maximum intensity of the time series.

## Statistical analysis

The error bars in the graphs represent the (*relative*) sample standard deviation. For each experiment, the sample size **n** is reported in the figure legend. 'Technical replicates' refers to in vitro assays performed on yeast lysates from a single batch on different days. 'Biological replicates' refers to assays performed on cells, which were cultured on different days from the DMSO-stocks from the same batch/passage number, then seeded and used in the experiment.

## Compound synthesis and characterization

### Synthesis of 1-(but-3-yn-1-yl)—4-nitrobenzene (2)

To a stirred solution of aldehyde **1** (141 mg, 0.787 mmol, 1.0 equiv.) in MeOH (10 mL) was added $K_2CO_3$ (217 mg, 1.57 mmol, 2.0 equiv.) followed by dimethyl(1-diazo-2-oxopropyl)phosphonate (*Müller et al., 1996*) (182 mg, 0.947 mmol, 1.2 equiv.) dropwise via syringe at ambient temperature. After stirring for 2 hr, pH 7 buffer was added to the dark red mixture. The aqueous phase was extracted with EtOAc (3 × 15 mL). The combined organic fractions were washed with water (15 mL) and brine (15 mL), then dried over anhydrous magnesium sulfate, filtered and concentrated under reduced pressure. The crude material was purified by flash column chromatography (65% $CH_2Cl_2$ in hexanes) to afford alkyne **2** as a pale yellow powder (132 mg, 0.753 mmol, 95%) (*McIntosh et al., 2012*; *MacHin and Pagenkopf, 2011*).

**TLC (65% $CH_2Cl_2$ in hexane): $R_f$: 0.77. [1]H NMR (CDCl$_3$, 400 MHz, 25°C): δ** 8.17 (d, *J* = 8.5 Hz, 2H), 7.41 (d, *J* = 8.5 Hz, 2H), 2.95 (t, *J* = 7.2 Hz, 2H), 2.55 (td, *J* = 7.2, 2.6 Hz, 2H), 2.01 (t, *J* = 2.6 Hz, 1H). **[13]C NMR (CDCl$_3$, 101 MHz, 25°C): δ** 148.01, 146.86, 129.53, 123.78, 82.64, 70.02, 34.52, 20.11. **IR (neat, ATR):** 3260, 3108, 3080, 2960, 2936, 2910, 2841, 1606, 1597, 1511, 1449, 1428, 1336, 1318, 1261, 1243, 1207, 1181, 1105, 1013, 944, 862, 848, 820, 805, 745, 697, 687, 675, 649, 631. **HRMS (EI[+]):** m/z calcd. for [$C_{10}H_9NO_2$][+]: 175.0633, found: 175.0627. **Melting point:** 110–111°C.

### Synthesis of 4-(but-3-yn-1-yl)aniline (3)

To a stirred solution of **2** (132 mg, 0.753 mmol) in EtOH (25 mL) was added $SnCl_2.2H_2O$ (1.70 g, 7.53 mmol, 10 equiv.). The mixture was heated to reflux and stirred for 19 hr. The reaction mixture was cooled to ambient temperature and 1M NaOH solution (50 mL) was added. The suspension was diluted with $CH_2Cl_2$ (100 mL) and filtered through a Celite plug, and the plug and precipitate were washed with $CH_2Cl_2$ (2 × 50 mL). The combined washings were separated and the organic layers were washed with brine (30 mL) and dried with anhydrous magnesium sulfate. The solution was subsequently filtered and concentrated under reduced pressure to yield alkyne **3** as a yellow oil (88.8 mg, 0.612 mmol, 82%) (*Li et al., 2013*).

**TLC ($CH_2Cl_2$): $R_f$: 0.35. [1]H NMR (CDCl$_3$, 400 MHz, 25°C): δ** 7.03 (d, *J* = 7.8 Hz, 2H), 6.64 (d, *J* = 7.7 Hz, 2H), 3.58 (s, 2H), 2.75 (t, *J* = 7.4 Hz, 2H), 2.45–2.42 (m, 2H), 1.99 (d, *J* = 1.5 Hz, 1H). **[13]C NMR (CDCl$_3$, 101 MHz, 25°C): δ** 144.79, 130.51, 129.26, 115.22, 84.27, 68.82, 34.08, 20.99. **IR (neat, ATR):** 3433, 3359, 3286, 3017, 2926, 2858, 2113, 1708, 1622, 1515, 1429, 1274, 1180, 1125, 1084, 822, 735, 629, 566. **HRMS (EI[+]):** m/z calcd. for [$C_{10}H_{11}N$][+]: 145.0892, found: 145.0882.

### Synthesis of 4-(4-((4-(but-3-yn-1-yl)phenyl)diazenyl)phenyl)butanoic acid (cFAAzo-4)

4-(4-Nitrosophenyl)butanoic acid was prepared by adding a solution of Oxone (510 mg, 1.67 mmol, 2.0 equiv.) in water (10 mL) to a stirring solution of 4-(4-aminophenyl)butyric acid (150 mg, 0.837 mmol, 1.0 equiv.) in $CH_2Cl_2$ (2.5 mL) at ambient temperature. The reaction mixture was stirred vigorously for 4 hr at room temperature, whereupon 2 M HCl (10 mL) was added and the aqueous layer was then extracted with $CH_2Cl_2$ (3 × 10 mL). The combined organic layers were washed with water (10 mL) and brine (10 mL), dried with anhydrous magnesium sulfate, then filtered and concentrated

under reduced pressure to afford 4-(4-nitrosophenyl)butanoic acid as a yellow powder, which was taken directly to the next reaction (*Priewisch and Rück-Braun, 2005*).

4-(4-Nitrosophenyl)butanoic acid (110 mg, 0.569 mmol) and alkyne **3** (89 mg, 0.61 mmol) were dissolved in AcOH (6 mL) and stirred at ambient temperature for 17 hr. Reaction progress was monitored by TLC. The solvent was removed under reduced pressure, and the crude material purified by flash column chromatography (10:89:1 → 25:74:1 EtOAc/hexanes/AcOH) to afford **cFAAzo-4** as a yellow gum (36 mg, 0.11 mmol, 20%).

**TLC (20:79:1 EtOAc/hexane/AcOH): $R_f$: 0.27.** **$^1$H NMR (CDCl$_3$, 400 MHz, 25°C): δ** 7.84 (dd, $J$ = 8.0, 5.5 Hz, 4H), 7.37 (d, $J$ = 8.2 Hz, 2H), 7.32 (d, $J$ = 8.2 Hz, 2H), 2.92 (t, $J$ = 7.5 Hz, 2H), 2.75 (t, $J$ = 7.6 Hz, 2H), 2.54 (td, $J$ = 7.4, 2.6 Hz, 2H), 2.41 (t, $J$ = 7.3 Hz, 2H), 2.03–2.00 (m, 3H). **$^{13}$C NMR (CDCl$_3$, 101 MHz, 25°C): δ** 178.48, 151.53, 151.38, 144.63, 143.66, 129.33, 129.32, 123.07, 123.01, 83.56, 69.39, 35.01, 34.79, 33.18, 26.18, 20.54. **IR (neat, ATR):** 3272, 2952, 2854, 1693, 1601, 1516, 1412, 1344, 1282, 1216, 1154, 911, 850. **HRMS (ESI$^+$):** m/z calcd. for $[C_{20}H_{22}N_2O_2]^+$: 321.1598, found: 321.1594 ([M + H]$^+$).

## Synthesis of 4-(4-(4-(but-3-yn-1-yl)phenyl)diazenyl)phenyl)-*N*-((2*S*,3*R*,*E*)−1,3-dihydroxyoctadec-4-en-2-yl)butanamide (caCer-1)

To a stirred solution of **cFAAzo-4** (12.0 mg, 0.0371 mmol, 1.0 equiv.) in EtOAc (1.5 mL) was added TBTU (12.0 mg, 0.0374 mmol, 1.0 equiv.) at ambient temperature. After stirring for 1 hr, D-*erythro*-sphingosine (14.6 mg, 0.0487 mmol, 1.3 equiv.) was added to the solution, followed by Et$_3$N (20.1 μL, 0.150 mmol, 4.0 equiv.). The reaction was continued for 15 hr, when progress was determined to be complete via TLC analysis. Aqueous saturated NaHCO$_3$ solution (3 mL) was added to the reaction mixture, and the aqueous layer was separated and further extracted with EtOAc (2 × 5 mL). The combined organic layers were washed with brine (5 mL), dried over anhydrous sodium sulfate, filtered and concentrated under reduced pressure. The residue was purified by flash column chromatography (75 → 85% EtOAc in hexanes) to afford **caCer-1** (14.8 mg, 0.0246 mmol, 66%) as an orange powder.

**TLC (80% EtOAc in hexane): $R_f$: 0.22.** **$^1$H NMR (CDCl$_3$, 400 MHz, 25°C): δ** 7.83 (t, $J$ = 8.2 Hz, 4H), 7.34 (dd, $J$ = 17.0, 8.5 Hz, 4H), 6.28 (d, $J$ = 7.6 Hz, 1H), 5.81–5.75 (m, 1H), 5.55–5.49 (m, 1H), 4.30–4.28 (m, 1H), 3.97–3.89 (m, 2H), 3.68 (dd, $J$ = 11.1, 3.2 Hz, 1H), 2.92 (t, $J$ = 7.4 Hz, 2H), 2.74 (t, $J$ = 7.5 Hz, 2H), 2.54 (td, $J$ = 7.5, 2.6 Hz, 2H), 2.28–2.24 (m, 2H), 2.05–1.99 (m, 5H), 1.34–1.23 (m, 24H), 0.89–0.86 (m, 3H). **$^{13}$C NMR (CDCl$_3$, 101 MHz, 25°C): δ** 173.32, 151.47, 151.33, 144.87, 143.67, 134.42, 129.36, 129.31, 128.83, 123.03, 123.00, 83.53, 74.79, 69.39, 62.50, 54.45, 35.89, 35.16, 34.77, 32.43, 32.06, 29.83, 29.80, 29.77, 29.63, 29.51, 29.37, 29.25, 27.00, 22.84, 20.52, 14.28. **IR (neat, ATR):** 3294, 2919, 2850, 1646, 1603, 1547, 1498, 1466, 1417, 1378, 1302, 1271, 1223, 1201, 1154, 1102, 1056, 1025, 1013, 961, 920, 892, 850, 832, 720, 633, 571, 560. **HRMS (ESI$^+$):** m/z calcd. for $[C_{38}H_{56}N_3O_3]^+$: 602.4316, found: 602.4316 ([M + H]$^+$). **Melting point:** 103–105°C.

## Synthesis of 4-((4-(but-3-yn-1-yl)phenyl)diazenyl)benzoic acid (cFAAzo-1)

Methyl 4-nitrosobenzoate was prepared by adding a solution of Oxone (1.50 g, 4.91 mmol, 5.0 equiv.) in water (40 mL) to a stirred solution of 4-carboxymethyl aniline (150 mg, 0.992 mmol, 1.0 equiv.) in CH$_2$Cl$_2$ (10 mL). The reaction mixture was stirred vigorously for 4 hr at room temperature. The reaction was quenched with 2 M HCl (50 mL) and then extracted with CH$_2$Cl$_2$ (3 × 50 mL). The combined organic layers were washed with water and brine, dried with anhydrous magnesium sulfate, then filtered and concentrated under reduced pressure to afford methyl 4-nitrosobenzoate as a yellow powder, which was used without further purification (*Priewisch and Rück-Braun, 2005*; *Nishioka et al., 2007*).

Methyl 4-nitrosobenzoate (43 mg, 0.26 mmol, 1.5 equiv.) and alkyne **3** (25 mg, 0.17 mmol, 1.0 equiv.) were dissolved in AcOH (2 mL) and stirred at ambient temperature for 18 hr. Reaction progress was monitored by TLC. The reaction solvent was removed under reduced pressure. The crude material was then eluted through a silica gel plug (25% EtOAc in hexanes) and the solvent removed under reduced pressure to afford the product mixture as a yellow powder, which could more conveniently be purified after subsequent ester cleavage.

The crude carbomethoxyaryl mixture was dissolved in MeOH (2 mL) and 2 M NaOH (0.4 mL) was added. The reaction was stirred for 16 hr at ambient temperature and was then acidified by the addition of 1 M HCl solution (1 mL). The mixture was extracted with EtOAc, then washed with water and brine. The organic layer was dried with anhydrous magnesium sulfate, then filtered and concentrated under reduced pressure and the crude material was purified by flash column chromatography (silica gel, 10:89:1 → 25:74:1 EtOAc/hexane/AcOH) to afford **cFAAzo-1** as a pale-yellow powder (16 mg, 0.057 mmol, 34% over two steps).

**TLC (25:74:1 EtOAc/hexane/AcOH):** $R_f$: 0.30. **$^1$H NMR (acetone-$d_6$, 400 MHz, 25°C):** δ 8.26 (d, $J$ = 8.6 Hz, 2H), 8.02 (d, $J$ = 8.6 Hz, 2H), 7.95 (d, $J$ = 8.4 Hz, 2H), 7.55 (d, $J$ = 8.5 Hz, 2H), 2.97 (t, $J$ = 7.3 Hz, 2H), 2.59 (td, $J$ = 7.4, 2.7 Hz, 2H), 2.41 (t, $J$ = 2.7 Hz, 1H). **$^{13}$C NMR (acetone-$d^6$, 101 MHz, 25°C):** δ 166.98, 155.97, 152.06, 146.21, 133.23, 131.65, 130.46, 123.88, 123.35, 83.99, 70.79, 35.26, 20.62. **IR (neat, ATR):** 3281, 2931, 2362, 1686, 1603, 1544, 1503, 1427, 1310, 1292, 1142, 1098, 1012, 946, 867, 835, 778, 692. **HRMS (EI$^+$):** m/z calcd. for $[C_{17}H_{14}N_2O_2]^+$: 278.1055, found: 278.1047. **Melting point:** 235°C (decomposed).

## Synthesis of 4-(4-(but-3-yn-1-yl)phenyl)diazenyl-N-((2S,3R,E)−1,3-dihydroxyoctadec-4-en-2-yl)benzamide (caCer-2)

To a stirred solution of **cFAAzo-1** (4.2 mg, 0.015 mmol, 1.0 equiv.) in EtOAc/DMF (5:1, 0.6 mL) was added HBTU (5.7 mg, 0.015 mmol,1.0 equiv.) under an inert atmosphere at ambient temperature. After 1 hr, D-*erythro*-sphingosine (5.4 mg, 0.018 mmol, 1.2 equiv.) was added, followed by Et$_3$N (6.3 µL, 0.045 mmol, 3.0 equiv.). The reaction was continued for 6.5 hr, when progress was determined to be complete via TLC. The reaction was quenched by the addition of saturated NaHCO$_3$ solution (5 mL). The mixture was extracted with EtOAc (3 × 5 mL), and the combined organic layers were washed with brine (5 mL), dried with anhydrous sodium sulfate, filtered and concentrated under reduced pressure. The crude product was purified by flash column chromatography (60 → 75% EtOAc in pentane) to afford **caCer-2** as an orange powder (7.0 mg, 0.013 mmol, 83%).

**TLC (75% EtOAc in hexane):** $R_f$: 0.43. **$^1$H NMR (CDCl$_3$, 400 MHz, 25°C):** δ 7.92 (d, $J$ = 1.0 Hz, 4H), 7.88 (d, $J$ = 8.4 Hz, 2H), 7.38 (d, $J$ = 8.4 Hz, 2H), 7.12 (d, $J$ = 7.6 Hz, 1H), 5.87–5.80 (m, 1H), 5.60 (dd, $J$ = 15.4, 6.3 Hz, 1H), 4.47 (t, $J$ = 4.7 Hz, 1H), 4.14–4.07 (m, 2H), 3.83 (dd, $J$ = 11.2, 3.1 Hz, 1H), 2.93 (d, $J$ = 14.8 Hz, 2H), 2.54 (td, $J$ = 7.4, 2.6 Hz, 2H), 2.06 (dt, $J$ = 11.0, 5.3 Hz, 2H), 2.01 (t, $J$ = 4.1 Hz, 1H), 1.37–1.34 (m, 2H), 1.23 (s, 22H), 0.87 (t, $J$ = 6.9 Hz, 3H). **$^{13}$C NMR (CDCl$_3$, 101 MHz, 25°C):** δ 167.19, 154.59, 151.39, 144.62, 135.85, 134.71, 129.45, 128.88, 128.21, 123.39, 123.03, 83.43, 75.02, 69.48, 62.52, 54.88, 34.81, 32.46, 32.08, 29.85, 29.84, 29.81, 29.76, 29.64, 29.52, 29.36, 29.27, 22.85, 20.47, 14.29. **IR (neat, ATR):** 3295, 2921, 2851, 1637, 1541, 1493, 1467, 1341, 1297, 1055, 1014, 964, 858. **HRMS (ESI$^+$):** m/z calcd. for $[C_{35}H_{50}N_3O_3]^+$: 560.3847, found: 560.3851 ($[M + H]^+$). **Melting point:** 120°C.

## Synthesis of pentadec-7-yn-1-ol (S1)

A solution of 1-nonyne (0.500 g, 4.03 mmol, 1.0 equiv.) in THF (30 mL) and HMPA (8 mL) was cooled to –78°C and treated with a solution of *n*–BuLi (2.48 M in hexanes, 3.41 mL, 8.45 mmol, 2.1 equiv.). The opaque, black solution was warmed to 0°C and stirred for 1 hr, then cooled once more to –78°C whereupon 6-bromohexanoic acid (0.785 g, 4.03 mmol, 1.0 equiv.) in THF (6 mL) was added dropwise. The reaction mixture was allowed to warm to ambient temperature and stirred for 48 hr, where the solution became clear and pale brown. Saturated aqueous ammonium chloride solution (25 mL) was then added to the solution and the aqueous layer was separated and extracted with EtOAc (3 × 30 mL). The combined organic layers were washed with distilled water (2 × 20 mL), saturated aqueous lithium chloride solution (2 × 20 mL) and brine (20 mL), dried over anhydrous sodium sulfate, filtered and concentrated. The pale orange oil was used directly in the next procedure without purification.

The crude oil from the previous step was dissolved in THF (5 mL) and was added dropwise to a suspension of lithium aluminum hydride (229 mg, 6.05 mmol, 1.5 equiv.) cooled to 0°C with an ice bath. The mixture was allowed to warm to room temperature and stirred for 1.5 hr, where thin layer chromatography analysis indicated complete conversion. The mixture was cooled with an ice bath and carefully quenched with distilled water (5 mL) followed by 2 M aqueous sodium hydroxide solution (10 mL). The aqueous phase was extracted with Et$_2$O (3 × 30 mL) and the combined organic

layers were washed with saturated aqueous ammonium chloride (30 mL) and brine (20 mL), dried over anhydrous sodium sulfate, filtered and concentrated under reduced pressure. The crude oil was purified by flash column chromatography (15% EtOAc in pentane) to yield pentadec-7-yn-1-ol **S1** (127 mg, 0.567 mmol, 14% over two steps) as a colorless oil.

**TLC (30% EtOAc in pentane):** $R_f$=0.62. **$^1$H NMR (CDCl$_3$, 400 MHz, 25°C):** δ 3.64 (t, $J$ = 6.6 Hz, 2H), 2.19–2.08 (m, 4H), 1.58 (p, $J$ = 6.7 Hz, 2H), 1.53–1.18 (m, 17H), 0.88 (t, $J$ = 6.8 Hz, 3H). **$^{13}$C NMR (CDCl$_3$, 100 MHz, 25°C):** δ 80.55, 80.16, 63.14, 32.83, 31.93, 29.31, 29.21, 28.99, 28.99, 28.74, 25.43, 22.79, 18.90, 18.84, 14.26. **IR (neat, ATR):** 3328, 2927, 2856, 1460, 1434, 1378, 1332, 1073, 1054, 1030, 724. **MS (FAB$^+$):** m/z calcd. for $[C_{16}H_{17}N_2O]^+$: 225.4, found: 225.4 $([M + H]^+)$.

## Synthesis of pentadec-14-yn-1-ol (S2)

Sodium hydride (60% in mineral oil, 149 mg, 3.74 mmol, 8.0 equiv.) was added in one portion to 1,3-diaminopropane (5 mL) at room temperature. The mixture was heated to 70°C and stirred for 1 hr, where it became opaque and brown, then cooled to room temperature. Pentadec-7-yn-1-ol **S1** (105 mg, 0.467 mg, 1.0 equiv.) in 1,3-diaminopropane (2 mL) was added to the vessel and the mixture was heated to 60°C and stirred for 19 hr. The reaction mixture was allowed to cool to room temperature and diluted with Et$_2$O (10 mL). Distilled water (10 mL) was then carefully added and the aqueous layer was separated and further extracted with Et$_2$O (3 × 10 mL). The combined organic layers were washed with distilled water (10 mL), 1 M HCl (10 mL) and brine (10 mL), dried over anhydrous sodium sulfate, filtered and concentrated. The crude residue was purified by flash column chromatography (15% EtOAc in pentane) to yield pentadec-14-yn-1-ol **S2** (70.3 mg, 0.313 mmol, 67%) as a white solid. The $^1$H NMR spectrum is in agreement with that previously reported (*Oppolzer et al., 2001*).

**TLC (20% EtOAc in pentane):** $R_f$=0.44. **$^1$H NMR (CDCl$_3$, 400 MHz, 25°C):** δ 3.61 (t, $J$ = 6.7 Hz, 2H), 2.16 (td, $J$ = 7.1, 2.7 Hz, 2H), 1.92 (t, $J$ = 2.7 Hz, 1H), 1.59–1.46 (m, 5H), 1.40–1.22 (m, 18H).

## Synthesis of pentadec-14-ynoic acid (4)

A chromic acid oxidizing solution was prepared according to literature (*Eisenbraun, 1965*). Chromium trioxide (67.0 g, 670 mmol) was dissolved in distilled water (125 mL) and the solution was cooled to 0°C. Concentrated sulfuric acid (58 mL) was added to the solution and the mixture was allowed to warm to room temperature. Distilled water was added to bring the total volume of the solution to 225 mL, making a ~ 3 M aqueous solution of chromic acid oxidizing solution. The chromic acid solution (0.149 mL, 0.446 mmol, 1.5 eq.) was added dropwise to a solution of pentadec-14-yn-1-ol **S2** (66.7 mg, 0.297 mmol, 1.0 equiv.) in acetone (2.0 mL) at 0°C. The mixture was stirred for 2 hr at 0°C until TLC analysis indicated complete conversion. The solution was filtered through a pad of Celite and the vessel and pad were washed with Et$_2$O (50 mL). The filtrate was washed with distilled water (20 mL) and brine (20 mL) and concentrated under reduced pressure. The crude residue was purified by flash column chromatography (10 → 30% EtOAc in pentane) to yield pentadec-14-ynoic acid **4** (58.1 mg, 0.244 mmol, 82%) as a waxy white solid. The $^1$H NMR spectrum is in agreement with that previously reported (*Seike et al., 2006*).

**TLC (20% EtOAc in pentane):** $R_f$=0.24. **$^1$H NMR (CDCl$_3$, 400 MHz, 25°C):** δ 11.24 (broad s, 1H), 2.34 (t, $J$ = 7.5 Hz, 2H), 2.18 (td, $J$ = 7.1, 2.7 Hz, 2H), 1.94 (t, $J$ = 2.6 Hz, 1H), 1.62 (q, $J$ = 7.2 Hz, 2H), 1.52 (p, $J$ = 7.2 Hz, 2H), 1.43–1.19 (m, 16H).

## Synthesis of 4-((4-propylphenyl)diazenyl)benzyl alcohol (5)

A solution of 4-propylaniline (2.50 g, 18.5 mmol, 2.1 equiv.) in CH$_2$Cl$_2$ (80 mL) was treated with Oxone (22.7 g, 74.0 mmol, 8.5 equiv.) in distilled water (100 mL) at room temperature and the biphasic mixture was stirred vigorously at room temperature for 20 hr. The aqueous phase was separated and further extracted with CH$_2$Cl$_2$ (2 × 60 mL). The combined organic phases were washed with 1 M hydrochloric acid solution (75 mL), saturated aqueous sodium bicarbonate solution (75 mL) and brine (75 mL), then dried over anhydrous sodium sulfate, filtered and concentrated under reduced pressure. The crude residue was purified by flash column chromatography (CH$_2$Cl$_2$), and the bright green fractions were collected, combined and concentrated to afford 4-propylnitrosobenzene as a clear green oil, which was taken directly to the next procedure.

The nitrosobenzene was redissolved in glacial acetic acid (75 mL) and 4-aminobenzyl alcohol (1.08 g, 8.74 mmol, 1.0 equiv.) in acetic acid (25 mL) was added to the solution at room temperature. The mixture was stirred vigorously for 72 hr at room temperature, then concentrated under reduced pressure and azeotroped twice with toluene (50 mL). The crude orange solid was purified by flash column chromatography (20 → 25% EtOAc in pentane) to afford 4-((4-propylphenyl)diazenyl)benzyl alcohol **5** (911 mg, 3.58 mmol, 41% yield) as an orange solid.

**TLC (30% EtOAc in pentane): R$_f$=0.43. $^1$H NMR (CDCl$_3$, 400 MHz, 25°C):** δ 7.90 (d, $J$ = 8.4 Hz, 2H), 7.85 (d, $J$ = 8.4 Hz, 2H), 7.50 (d, $J$ = 8.4 Hz, 2H), 7.32 (d, $J$ = 8.4 Hz, 2H), 4.77 (s, 2H), 2.67 (t, $J$ = 7.5 Hz, 2H), 1.89 (broad s, 1H), 1.70 (p, $J$ = 7.5 Hz, 2H), 0.97 (t, $J$ = 7.5 Hz, 3H).$^{13}$**C NMR (CDCl$_3$, 100 MHz, 25°C):** δ 152.32, 151.05, 146.49, 143.62, 129.31, 127.56, 123.07, 122.96, 65.03, 38.08, 24.55, 13.94. **IR (neat, ATR):** 3332, 2958, 2930, 2869, 1661, 1585, 1499, 1449, 1417, 1340, 1303, 1222, 1026, 1010, 852, 832. **HRMS (ESI$^+$):** m/z calcd. for [C$_{16}$H$_{19}$N$_2$O]$^+$: 255.1492, found: 255.1491 ([M + H]$^+$). **Melting point:** 138°C.

### Synthesis of 4-((4-propylphenyl)diazenyl)benzaldehyde (6)

A solution of 4-((4-propylphenyl)diazenyl)benzyl alcohol **5** (302 mg, 1.19 mmol, 1.0 equiv.) in CH$_2$Cl$_2$ (12 mL) was treated with Dess-Martin periodinane (655 mg, 1.55 mmol, 1.3 equiv.) at room temperature. The reaction mixture was left to stir for 45 min and a mixture of saturated aqueous sodium bicarbonate solution and saturated aqueous sodium thiosulfate solution (1:1, 20 mL) was then added. The biphasic mixture was stirred for 30 min, then the aqueous phase was separated and extracted with CH$_2$Cl$_2$ (2 × 20 mL). The combined organic layers were washed with saturated aqueous sodium bicarbonate solution (30 mL), dried over anhydrous sodium sulfate, filtered and concentrated under reduced pressure. The crude residue was purified by flash column chromatography (3 → 4% EtOAc in pentane) to afford 4-((4-propylphenyl)diazenyl)benzaldehyde **6** (282 mg, 1.12 mmol, 94%) as a red crystalline solid.

**TLC (30% EtOAc in pentane): R$_f$=0.43. $^1$H NMR (CDCl$_3$, 400 MHz, 25°C):** δ 10.07 (s, 1H), 8.00 (s, 4H), 7.88 (d, $J$ = 8.1 Hz, 2H), 7.32 (d, $J$ = 8.1 Hz, 2H), 2.66 (t, $J$ = 7.6 Hz, 2H), 1.69 (p, $J$ = 7.5 Hz, 2H), 0.97 (t, $J$ = 7.4 Hz, 3H).$^{13}$**C NMR (CDCl$_3$, 100 MHz, 25°C):** δ 191.64, 156.01, 150.91, 147.57, 137.24, 130.70, 129.34, 123.36, 123.25, 38.04, 24.40, 13.88. **IR (neat, ATR):** 3023, 2956, 2929, 2845, 2739, 1696, 1597, 1581, 1498, 1460, 1416, 1378, 1316, 1304, 1289, 1197, 1183, 1148, 1129, 1112, 1090, 1003, 908, 847, 831, 811, 793, 75, 729, 663. **HRMS (ESI$^+$):** m/z calcd. for [C$_{16}$H$_{19}$N$_2$O]$^+$: 253.1335, found: 253.1336 ([M + H]$^+$). **Melting point:** 107°C.

### Synthesis of 1-(4-propylphenyl)−2-(4-vinylphenyl)diazene (7)

A suspension of methyltriphenylphosphonium bromide (439 mg, 1.23 mmol, 1.1 equiv.) in THF (12 mL) at 0°C was treated with a solution of $n$-BuLi (2.48 m in hexanes, 0.496 mL, 1.23 mmol, 1.1 equiv.). The resulting bright yellow suspension was stirred for 20 min at 0°C and a solution of 4-((4-propylphenyl)diazenyl)benzaldehyde **6** (282 mg, 1.12 mmol, 1.0 equiv.) in THF (5 mL) was added dropwise. The mixture was allowed to warm to room temperature and stirred for 1 hr. Saturated aqueous ammonium chloride solution (20 mL) was added and the aqueous phase was extracted with CH$_2$Cl$_2$ (3 × 20 mL). The combined organic layers were dried over anhydrous sodium sulfate, filtered and concentrated under reduced pressure. The residue was purified by flash column chromatography (dry loading with 1 g silica gel, 0 → 3% Et$_2$O in pentane) to yield 1-(4-propylphenyl)−2-(4-vinylphenyl)diazene **7** (237 mg, 0.946 mmol, 84%) as a crystalline orange solid.

**TLC (30% EtOAc in pentane): R$_f$=0.43. $^1$H NMR (CDCl$_3$, 400 MHz, 25°C):** δ 7.89 (d, $J$ = 8.5 Hz, 2H), 7.86 (d, $J$ = 8.4 Hz, 2H), 7.55 (d, $J$ = 8.5 Hz, 2H), 7.33 (d, $J$ = 8.4 Hz, 2H), 6.79 (dd, $J$ = 17.6, 10.9 Hz, 1H), 5.87 (d, $J$ = 17.6 Hz, 1H), 5.36 (d, $J$ = 10.9 Hz, 1H), 2.68 (t, $J$ = 7.5 Hz, 2H), 1.70 (h, $J$ = 7.5 Hz, 2H), 0.98 (t, $J$ = 7.5 Hz, 3H). $^{13}$**C NMR (CDCl$_3$, 100 MHz, 25°C):** δ 152.31, 151.17, 146.42, 140.00, 136.33, 129.30, 127.03, 123.21, 122.96, 115.55, 38.09, 24.56, 13.95. **IR (neat, ATR):** 3045, 2958, 2929, 2870, 1626, 1598, 1497, 1454, 1414, 1402, 1303, 1287, 1226, 1155, 1109, 1011, 988, 908, 848, 802, 748. **HRMS (ESI$^+$):** m/z calcd. for [C$_{17}$H$_{19}$N$_2$]$^+$: 251.1543, found: 251.1543 ([M + H]$^+$). **Melting point:** 37°C.

## Synthesis of 4-iodo-4'-methylazobenzene (S3)

4-Iodo-4'-methylazobenzene was synthesized following a modified procedure of *Strueben et al. (2014)*. A solution of *p*-toluidine (2.50 g, 24.3 mmol, 1.0 equiv.) in $CH_2Cl_2$ (80 mL) was treated with Oxone (29.8 g, 97.1 mmol, 4.1 equiv.) in distilled water (120 mL) at room temperature and the biphasic mixture was stirred vigorously at room temperature for 18 hr. The aqueous phase was separated and further extracted with $CH_2Cl_2$ (2 × 50 mL). The combined organic phases washed with 1 M hydrochloric acid solution (80 mL), saturated aqueous sodium bicarbonate solution (80 mL) and brine (80 mL), then dried over anhydrous sodium sulfate, filtered and concentrated under reduced pressure. The crude residue was purified by flash column chromatography ($CH_2Cl_2$), and the bright green fractions were collected, combined and concentrated to afford a clear green oil. The oil was redissolved in $CH_2Cl_2$ (20 mL) and acetic acid (30 mL) and 4-iodoaniline (5.13 g, 23.4 mmol, 1.0 equiv.) was added to the solution. The mixture was stirred for 15 hr at room temperature, during which an orange-yellow crystalline solid precipitated. The mixture was concentrated under reduced pressure, suspended in ice-cold ethanol (30 mL) and filtered. The recovered crystals were washed with ice-cold ethanol (30 mL) and dried to afford 4-iodo-4'-methylazobenzene **S3** (3.35 g, 10.4 mmol, 45% yield) as an orange-yellow crystalline solid. The $^1$H NMR and $^{13}$C NMR spectra are in agreement with those previously reported (*Strueben et al., 2014*).

**$^1$H NMR (CDCl$_3$, 400 MHz, 25°C):** δ 7.85 (d, *J* = 8.5 Hz, 2H), 7.82 (d, *J* = 8.5 Hz, 2H), 7.64 (d, *J* = 8.5 Hz, 2H), 7.32 (d, *J* = 8.5 Hz, 3H), 2.44 (s, 3H). **$^{13}$C NMR (CDCl$_3$, 100 MHz, 25°C):** δ 152.13, 150.70, 142.17, 138.43, 129.96, 124.51, 123.12, 97.37, 21.72.

## Synthesis of 3-(4-(*p*-tolyldiazenyl)phenyl)propanal (S4)

1-(4-Iodophenyl)−2-(*p*-tolyl)diazene (3.34 g, 10.4 mmol, 1.0 equiv.) was suspended in DMF (12 mL) and toluene (12 mL) at room temperature and tetrabutylammonium chloride (2.89 g, 10.4 mmol, 1.0 equiv.), sodium bicarbonate (2.18 g, 26.0 mmol, 2.5 equiv.) and allyl alcohol (0.906 g, 15.6 mmol, 1.5 equiv.) were added sequentially to the stirring mixture. The orange suspension was stirred for 10 min at room temperature, whereupon PdCl$_2$ (0.369 mg, 2.08 mmol, 0.20 equiv.) was added to the flask. The bright red suspension was warmed to 45°C and stirred for 2.5 hr, then cooled back to room temperature and stirred for 48 hr. The reaction mixture was then diluted with EtOAc (125 mL) and washed successively with 1 M aqueous hydrochloric acid solution (50 mL), distilled water (4 × 50 mL) and brine (50 mL). The organic layer was dried over anhydrous sodium sulfate, filtered and concentrated. The crude residue was purified by flash column chromatography (6 → 10% EtOAc in pentane) to yield 3-(4-(*p*-tolyldiazenyl)phenyl)propanal (2.28 g, 9.07 mmol, 87%) as a crystalline orange solid.

**TLC (10% EtOAc in pentane): R$_f$=0.36. $^1$H NMR (CDCl$_3$, 400 MHz, 25°C):** δ 9.82 (s, 1H), 7.85 (d, *J* = 7.5 Hz, 2H), 7.83 (d, *J* = 7.5 Hz, 2H), 7.33 (d, *J* = 7.5 Hz, 2H), 7.31 (d, *J* = 7.5 Hz, 2H), 3.02 (t, *J* = 7.5 Hz, 2H), 2.81 (t, *J* = 7.5 Hz, 2H), 2.43 (s, 3H). **$^{13}$C NMR (CDCl$_3$, 100 MHz, 25°C):** δ 201.18, 151.36, 150.78, 143.48, 141.52, 129.80, 129.06, 123.06, 122.85, 45.08, 27.97, 21.57. **IR (neat, ATR):** 3022, 2921, 2832, 2730, 1716, 1600, 1497, 1448, 1416, 1390, 1356, 1302, 1221, 1210, 1153, 1111, 1061, 1038, 1011, 904, 843, 822, 728, 705, 682. **HRMS (ESI$^+$):** m/z calcd. for [C$_{16}$H$_{17}$N$_2$O]$^+$: 253.1335, found: 253.1336 ([M + H]$^+$). **Melting point:** 78°C.

## Synthesis of 1-(4-(but-3-en-1-yl)phenyl)−2-(*p*-tolyl)diazene (S5)

A suspension of methyltriphenylphosphonium bromide (3.87 g, 10.8 mmol, 1.2 equiv.) in THF (50 mL) at –78°C was treated with a solution of *n*-BuLi (2.48 m in hexanes, 4.37 mL, 10.8 mmol, 1.2 equiv.). The resulting bright yellow suspension was warmed to 0°C for 20 min, then cooled once more to –78°C. 3-(4-(*p*-Tolyldiazenyl)phenyl)propanal (2.28 g, 9.02 mmol, 1.0 equiv.) in THF (10 mL) was added dropwise and the mixture was allowed to warm to room temperature and stirred for 15 hr. Saturated aqueous ammonium chloride solution (60 mL) was added and the aqueous phase was separated and further extracted with CH$_2$Cl$_2$ (2 × 80 mL). The combined organic layers were dried over anhydrous sodium sulfate, filtered and concentrated under reduced pressure. The residue was purified by flash column chromatography (dry loading, 1 → 2% EtOAc in pentane) to yield 1-(4-(but-3-en-1-yl)phenyl)−2-(*p*-tolyl)diazene (2.06 g, 8.22 mmol, 91%) as a crystalline orange solid.

**TLC (2% EtOAc in pentane): R$_f$=0.30. $^1$H NMR (CDCl$_3$, 400 MHz, 25°C):** δ 7.78 (d, *J* = 8.2 Hz, 2H), 7.76 (d, *J* = 7.5 Hz, 2H), 7.25 (d, *J* = 8.2 Hz, 2H), 7.23 (d, *J* = 7.5 Hz, 2H), 5.80 (ddt, *J* = 16.9,

10.2, 6.6 Hz, 1H), 4.99 (dd, $J$ = 16.9, 1.7 Hz, 1H), 4.94 (dd, $J$ = 10.2, 1.7 Hz, 1H), 2.72 (t, $J$ = 7.5 Hz, 2H), 2.36 (m, 5H). $^{13}$C NMR (CDCl$_3$, 100 MHz, 25°C): δ 151.20, 150.92, 145.14, 141.35, 137.80, 129.82, 129.23, 122.88, 122.86, 115.38, 35.42, 35.37, 21.61. IR (neat, ATR): 3074, 3054, 3024, 2977, 2922, 2857, 1640, 1601, 1580, 1497, 1440, 1415, 1302, 1224, 1209, 1155, 1105, 1012, 996, 950, 907, 837, 823, 708, 643. HRMS (ESI$^+$): m/z calcd. for [C$_{17}$H$_{19}$N$_2$]$^+$: 251.1543, found: 251.1542 ([M + H]$^+$). Melting point: 57°C.

## Synthesis of N-Boc-(R)−1-((S)−2,2-dimethyloxazolidin-4-yl)prop-2-en-1-ol (8)

Vinyl magnesium bromide (1.0 M in THF, 22.7 mL, 22.7 mmol, 2.0 equiv.) was added over 30 min via drop funnel to a solution of (S)−1,1-dimethylethyl 4-formyl-2,2-dimethyloxazolidine-3-carboxylate (2.61 g, 11.4 mmol, 1.0 equiv.) in THF (50 mL) at –78°C. The reaction mixture was stirred for 2 hr at – 78°C until TLC analysis indicated complete conversion, and saturated aqueous ammonium chloride solution (40 mL) was added at this temperature. After warming to room temperature, the aqueous layer was separated and extracted with EtOAc (3 × 70 mL). The combined organic layers were dried over anhydrous sodium sulfate, filtered and concentrated. The crude residue was purified by flash column chromatography (15% EtOAc in pentane) to afford N-Boc-(R)−1-((S)−2,2-dimethyloxazolidin-4-yl)prop-2-en-1-ol 8 (2.33 g, 9.07 mmol, 80%) as a colorless oil. High-temperature $^1$H NMR analysis indicates a 5.2:1 anti/syn mixture of diastereomers, consistent with previous literature results (Ojima and Vidal, 1998). Further purification of the mixture by careful column chromatography (10% EtOAc in pentane) yielded the pure anti diastereomer.

TLC (20% EtOAc in pentane): R$_f$=0.35. $^1$H NMR (toluene-d$_8$, 400 MHz, 90°C): δ 5.81 (ddd, $J$ = 16.7, 10.5, 5.2 Hz, 1H), 5.31 (dt, $J$ = 16.7, 1.8 Hz, 1H), 5.06 (d, $J$ = 10.5 Hz, 1H), 4.26 (broad s, 1H), 3.87 (m, 1H), 3.77 (m, 1H), 3.66 (dd, $J$ = 9.0, 6.8 Hz, 1H), 1.58 (s, 3H), 1.43 (s, 3H), 1.38 (s, 9H). $^{13}$C NMR (toluene-d$_8$, 100 MHz, 90°C): δ 153.39, 138.68, 115.44, 94.68, 80.27, 73.87, 64.81, 62.47, 28.49, 26.82, 24.43. IR (neat, ATR): 3461, 2978, 2936, 2879, 1694, 1478, 1456, 1377, 1365, 1255, 1206, 1170, 1095, 1049, 989, 923, 848, 807, 767. HRMS (ESI$^+$): m/z calcd. for [C$_{13}$H$_{24}$NO$_4$]$^+$: 258.1700, found: 258.1699 ([M + H]$^+$).

## Synthesis of N-Boc-(R,E)−1-((S)−2,2-dimethyloxazolidin-4-yl)−3-((4-(4-propylphenyl)diazenyl)phenyl)prop-2-en-1-ol (9)

Hoyveda-Grubbs 2$^{nd}$ generation catalyst (29.6 mg, 0.0472 mmol, 0.10 equiv.) was added to a solution of allyl alcohol 8 (121 mg, 0.472 mmol, 1.0 equiv.) and olefin 7 (236 mg, 0.944 mmol, 2.0 equiv.) in degassed CH$_2$Cl$_2$ (6 mL) at room temperature. The deep red mixture was heated to 45°C and stirred for 21 hr. The mixture was cooled to room temperature and, without concentrating, was loaded on an equilibrated silica gel column and purified by flash column chromatography (15% to 25% EtOAc in pentane). The combined fractions contained traces of remnant catalyst, and the product was purified a second time by flash column chromatography (15 → 25% EtOAc in pentane) to yield 9 (71.1 mg, 0.148 mmol, 31%) as a dark orange gum.

TLC (20% EtOAc in pentane): R$_f$=0.32. $^1$H NMR (toluene-d$_8$, 400 MHz, 90°C): δ 7.90 (d, $J$ = 8.3 Hz, 2H), 7.88 (d, $J$ = 8.3 Hz, 2H), 7.35 (d, $J$ = 8.4 Hz, 2H), 7.07 (d, $J$ = 8.3 Hz, 2H), 6.68 (dd, $J$ = 15.8, 1.5 Hz, 1H), 6.26 (dd, $J$ = 15.8, 5.6 Hz, 1H), 4.37 (m, 1H), 3.98 (m, 1H), 3.81 (m, 1H), 3.70 (dd, $J$ = 9.1, 6.7 Hz, 1H), 2.43 (t, $J$ = 7.3, 2H), 1.56 (s, 3H), 1.52 (q, $J$ = 7.3, 2H), 1.41 (s, 3H), 1.32 (s, 9H), 0.84 (t, $J$ = 7.3 Hz, 3H). $^{13}$C NMR (toluene-d$_8$, 100 MHz, 90°C): δ 153.79, 152.91, 152.14, 146.11, 140.24, 131.86, 130.52, 129.35, 127.55, 123.72, 123.49, 94.78, 80.53, 74.33, 65.21, 62.93, 38.27, 28.46, 27.02, 24.51, 13.81. IR (neat, ATR): 3426, 2976, 2933, 2873, 1693, 1600, 1497. 1477, 1455, 1389, 1376, 1366, 1255, 1205, 1156, 1100, 1068, 1050, 967, 921, 864, 847, 768, 733. HRMS (ESI$^+$): m/z calcd. for [C$_{28}$H$_{38}$N$_3$O$_4$]$^+$: 480.2857, found: 480.2860 ([M + H]$^+$).

## Synthesis of (2S,3R,E)−2-amino-5-(4-((4-propylphenyl)diazenyl)phenyl) pent-4-ene-1,3-diol (aSph-1)

(R,E)−1-((S)−2,2-dimethyloxazolidin-4-yl)−6-(4-(p-tolyldiazenyl)phenyl)hex-2-en-1-ol 9 (23.9 mg, 0.0498 mmol, 1.0 equiv.) was dissolved in THF (1.25 mL) and the solution was cooled to 0°C with an ice bath. 2 M hydrochloric acid (0.50 mL) was added dropwise and the reaction mixture was heated to 60°C. After stirring for 3 hr, the solution was cooled to ambient temperature and saturated sodium carbonate solution (5 mL) was added. The aqueous phase was extracted with CH$_2$Cl$_2$ (3 × 10

mL) and the combined organic layers were dried over anhydrous sodium sulfate, filtered and concentrated under reduced pressure. The crude orange solid was purified by flash column chromatography (5/94.5/0.5 → 10/89/1 MeOH/CH$_2$Cl$_2$/aqueous ammonium hydroxide solution) to afford **aSph-1** (13.1 mg, 0.0386 mmol, 77%) as an orange solid.

**TLC (40% acetone in toluene): R$_f$=0.14.** **$^1$H NMR (methanol-d$_4$, 400 MHz, 25°C): δ** 7.77 (d, J = 8.6 Hz, 2H), 7.73 (d, J = 8.4 Hz, 2H), 7.53 (d, J = 8.6 Hz, 2H), 7.26 (d, J = 8.4 Hz, 2H), 6.66 (d, J = 15.9 Hz, 1H), 6.38 (dd, J = 15.9, 6.8 Hz, 1H), 4.19 (t, J = 5.8 Hz, 1H), 3.65 (dd, J = 10.9, 4.6 Hz, 1H), 3.48 (dd, J = 10.9, 6.9 Hz, 1H), 2.85 (q, J = 5.7 Hz, 1H), 2.58 (d, J = 7.5 Hz, 2H), 1.60 (h, J = 7.4 Hz, 2H), 0.88 (t, J = 7.4 Hz, 3H).**$^{13}$C NMR (methanol-d$_4$, 100 MHz, 25°C): δ** 153.28, 152.36, 147.76, 141.00, 132.46, 132.26, 130.31, 128.38, 124.10, 123.82, 74.80, 64.24, 58.28, 38.89, 25.61, 14.10. **IR (neat, ATR):** 3045, 2958, 2929, 2870, 1626, 1598, 1497, 1454, 1414, 1402, 1303, 1287, 1226, 1155, 1109, 1011, 988, 908, 848, 802, 748. **HRMS (ESI$^+$):** m/z calcd. for [C$_{20}$H$_{26}$N$_3$O$_2$]$^+$: 340.2020, found: 340.2020 ([M + H]$^+$). **Melting point:** 135°C.

## Synthesis of N-((2S,3R,E)−1,3-dihydroxy-5-(4-((4-propylphenyl) diazenyl) phenyl)pent-4-en-2-yl)pentadec-14-ynamide (caCer-3)

**aSph-1** (7.9 mg, 0.023 mmol, 1.0 equiv.) and pentadec-14-ynoic acid **4** (6.1 mg, 0.026 mmol, 1.1 equiv.) were dissolved in CH$_2$Cl$_2$ (0.8 mL) and the solution was cooled to 0°C with an ice bath. Diisopropylethylamine (12.2 μL, 0.0698 mmol, 3.0 equiv.), 1-ethyl-3-(3-dimethylaminopropyl)carbodiimide hydrochloride (7.6 mg, 0.040 mmol, 1.7 equiv.) and 1-hydroxybenzotriazole hydrate (6.8 mg, 0.044 mmol, 1.9 equiv.) were added sequentially to the flask, and the mixture was allowed warm to room temperature and was stirred for 20 hr. Saturated aqueous sodium bicarbonate solution (5 mL) was then added to the mixture. The aqueous layer was separated and extracted with CH$_2$Cl$_2$ (3 × 5 mL). The combined organic layers were washed with 1 M hydrochloric acid (5 mL) and brine (5 mL), dried over anhydrous sodium sulfate, filtered and concentrated under reduced pressure. The residue was purified by flash column chromatography (2 → 4% MeOH in CH$_2$Cl$_2$) to afford **caCer-3** (11.0 mg, 0.0197 mmol, 85%) as an orange solid.

### Note

Due to initially proceeding with a mixture of epimers from compound **8**, our first synthesis of **caCer-3** produced an inseparable 7.3:1 mixture of *erythro/threo* ceramide by $^1$H NMR analysis and biological assays were conducted with this mixture. The procedures reported here are from a second synthesis conducted with *anti*-**8** to yield pure *erythro*-**caCer-3**.

**TLC (90% EtOAc in pentane): R$_f$=0.36.** **$^1$H NMR (CDCl$_3$, 400 MHz, 25°C): δ** 7.86 (d, J = 8.5 Hz, 2H), 7.83 (d, J = 8.3 Hz, 2H), 7.50 (d, J = 8.5 Hz, 2H), 7.31 (d, J = 8.3 Hz, 2H), 6.77 (d, J = 16.1 Hz, 1H), 6.38 (dd, J = 16.1, 5.7 Hz, 1H), 6.37 (s, 1H), 4.59 (t, J = 4.0 Hz, 1H), 4.05 (m, 2H), 3.78 (m, 1H), 2.66 (t, J = 7.5 Hz, 2H), 2.24 (m, 2H), 2.16 (td, J = 7.1, 2.6 Hz, 2H), 1.94 (t, J = 2.6 Hz, 1H), 1.66 (m, 4H), 1.50 (p, J = 7.3 Hz, 2H) 1.41–1.15 (m, 18H), 0.97 (t, J = 7.3 Hz, 3H). **$^{13}$C NMR (CDCl$_3$, 100 MHz, 25°C): δ** 174.30, 152.30, 151.11, 146.52, 138.71, 131.21, 130.19, 129.31, 127.38, 123.33, 122.99, 84.99, 74.62, 68.19, 62.52, 54.62, 38.09, 36.99, 29.74, 29.71, 29.64, 29.61, 29.52, 29.43, 29.25, 28.90, 28.62, 25.94, 24.55, 18.54, 13.95. **IR (neat, ATR):** 3296, 2922, 2850, 1645, 1600, 1547, 1467, 1440, 1302, 1257, 1155, 1116, 1065, 1002, 964, 862, 825, 724, 696. **HRMS (ESI$^+$):** m/z calcd. for [C$_{35}$H$_{50}$N$_3$O$_3$]$^+$: 560.3847, found: 560.3862 ([M + H]$^+$). **Melting point:** 133°C.

## Synthesis of N-Boc-(R,E)−1-((S)−2,2-dimethyloxazolidin-4-yl)−6-(4-(p-tolyldiazenyl)phenyl)hex-2-en-1-ol (S6)

Hoveyda-Grubbs catalyst, 2$^{nd}$ generation (63.8 mg, 0.102 mmol, 0.10 equiv.) was added to a solution of allyl alcohol **S5** (262 mg, 1.02 mmol, 1.0 equiv.) and 1-(4-(but-3-en-1-yl)phenyl)−2-(p-tolyl)diazene (510 mg, 2.04 mmol, 2.0 equiv.) in degassed CH$_2$Cl$_2$ (8 mL) at room temperature. The deep red mixture was heated to 45°C and stirred for 16 hr. The mixture was cooled to ambient temperature and, without concentrating, was loaded on an equilibrated silica gel column and purified by flash column chromatography (15 → 25% EtOAc in pentane). The combined fractions contained traces of remnant catalyst, and the product was purified a second time by flash column chromatography (15 → 25% EtOAc in pentane) to yield **S6** (223 mg, 0.450 mmol, 44%) as an orange gum.

TLC (20% EtOAc in pentane): $R_f$=0.35. $^1$H NMR (toluene-$d_8$, 400 MHz, 90°C): δ 7.89 (d, $J$ = 8.1 Hz, 2H), 7.86 (d, $J$ = 8.1 Hz, 2H), 7.08 (m, 4H), 5.72 (dt, $J$ = 13.3, 6.5 Hz, 1H), 5.49 (dd, $J$ = 15.4, 5.6 Hz, 1H), 4.26 (broad s, 1H), 3.87 (m, 1H), 3.77 (m, 1H), 3.66 (dd, $J$ = 9.0, 6.8 Hz, 1H), 2.62 (t, $J$ = 7.7 Hz, 2H), 2.30 (q, $J$ = 7.3 Hz, 2H), 2.15 (s, 3H), 1.58 (s, 3H), 1.43 (s, 3H), 1.38 (s, 9H). $^{13}$C NMR (toluene-$d_8$, 100 MHz, 90°C): 129.94, 129.37, 123.45, 123.38, 94.71, 80.24, 73.64, 64.92, 62.81, 35.99, 34.20, 28.60, 26.94, 24.55, 21.22. IR (neat, ATR): 3448, 2978, 2933, 1694, 1602, 1498, 1478, 1454, 1388, 1376, 1365, 1255, 1206, 1156, 1101, 1067, 1014, 968, 912, 842, 768, 733. HRMS (ESI$^+$): m/z calcd. for $[C_{28}H_{38}N_3O_4]^+$: 480.2857, found: 480.2873 ([M + H]$^+$).

## Synthesis of (2S,3R,E)−2-amino-7-(4-(p-tolyldiazenyl)phenyl)hept-4-ene-1,3-diol (aSph-2)

2 M Hydrochloric acid solution (2 mL) was added dropwise to a solution of **S6** (94.2 mg, 0.196 mmol, 1.0 equiv.) in THF (4 mL) at room temperature and the reaction mixture was heated to 60°C for 3 hr. The solution was then basified to pH 10 with 2 M sodium hydroxide solution (2.4 mL) and the aqueous mixture was extracted with $CH_2Cl_2$ (3 × 10 mL). The combined organic layers were washed with brine (10 mL), dried over anhydrous sodium sulfate, filtered and concentrated under reduced pressure. The crude orange solid was purified by flash column chromatography (5/94.5/0.5 → 10/89/1 MeOH/$CH_2Cl_2$/aqueous 25% ammonium hydroxide solution) to afford **aSph-2** (55.5 mg, 0.164 mmol, 84%) as an orange solid.

TLC (40% acetone in toluene): $R_f$=0.14. $^1$H NMR (methanol-$d_4$, 400 MHz, 25°C): δ 7.79 (d, $J$ = 8.1 Hz, 2H), 7.76 (d, $J$ = 7.9 Hz, 2H), 7.34 (d, $J$ = 8.1 Hz, 2H), 7.31 (d, $J$ = 7.9 Hz, 2H), 5.76 (dt, $J$ = 15.4, 6.8 Hz, 1H), 5.47 (dd, $J$ = 15.4, 6.8 Hz, 1H), 3.95 (t, $J$ = 6.7 Hz, 1H), 3.59 (dd, $J$ = 11.2, 4.1 Hz, 1H), 3.43 (dd, $J$ = 11.2, 6.9 Hz, 1H), 2.80 (q, $J$ = 6.8 Hz, 2H), 2.71 (q, $J$ = 6.0 Hz, 1H), 2.45 (m, 2H), 2.40 (s, 3H). $^{13}$C NMR (methanol-$d_4$, 100 MHz, 25°C): δ 152.39, 152.09, 146.59, 142.86, 133.97, 131.66, 130.80, 130.43, 123.77, 123.73, 74.42, 63.66, 57.95, 36.28, 35.05, 21.45. IR (neat, ATR): 3345, 3286, 3024, 2922, 2855, 1601, 1581, 1497, 1451, 1416, 1302, 1154, 1050, 1034, 1012, 965, 849, 827, 712, 643, 617, 558. HRMS (ESI$^+$): m/z calcd. for $[C_{20}H_{26}N_3O_2]^+$: 340.2020, found: 340.2022 ([M + H]$^+$). Melting point: 141°C.

## Synthesis of N-((2S,3R,E)−1,3-dihydroxy-7-(4-(p-tolyldiazenyl) phenyl)hept-4-en-2-yl)pentadec-14-ynamide (caCer-4)

**aSph-2** (15.0 mg, 0.0442 mmol, 1.0 equiv.) and pentadec-14-ynoic acid (10.5 mg, 0.0442 mmol, 1.0 equiv.) were dissolved in $CH_2Cl_2$ (1.5 mL) and the solution was cooled to 0°C with an ice bath. Diisopropylethylamine (23.1 μL, 0.133 mmol, 3.0 equiv.), 1-ethyl-3-(3-dimethylaminopropyl)carbodiimide hydrochloride (12.7 mg, 0.0663 mmol, 1.5 equiv.) and 1-hydroxybenzotriazole hydrate (11.5 mg, 0.0751 mmol, 1.7 equiv.) were added sequentially to the flask, and the mixture was allowed to warm to room temperature and was stirred for 4 hr. TLC analysis indicated some **aSph-2** remained, and the mixture was cooled to 0°C and additional pentadec-14-ynoic acid (5.2 mg, 0.022 mmol, 0.5 equiv.), diisopropylethylamine (7.7 μL, 0.044 mmol, 1.0 equiv.), 1-ethyl-3-(3-dimethylaminopropyl) carbodiimidehydrochloride (4.2 mg, 0.022 mmol, 0.5 equiv.) and 1-hydroxybenzotriazole hydrate (3.4 mg, 0.22 mmol, 0.5 equiv.) were added. The mixture was left to stir at room temperature for 17 hr and saturated aqueous sodium bicarbonate solution (5 mL) was added. The aqueous layer was separated and extracted with $CH_2Cl_2$ (3 × 5 mL). The combined organic layers were washed with 1 M hydrochloric acid (5 mL) and brine (5 mL), dried over anhydrous sodium sulfate, filtered and concentrated under reduced pressure. The residue was purified by flash column chromatography (80% EtOAc in pentane) to afford **caCer-4** (17.2 mg, 0.0307 mmol, 70%) as an orange solid.

TLC (80% EtOAc in pentane): $R_f$=0.30. $^1$H NMR (CDCl$_3$, 400 MHz, 25°C): δ 7.82 (d, $J$ = 8.4 Hz, 2H), 7.80 (d, $J$ = 8.4 Hz, 2H), 7.31 (d, $J$ = 8.4 Hz, 2H), 7.29 (d, $J$ = 8.4 Hz, 2H), 6.20 (d, $J$ = 7.6 Hz, 1H), 5.79 (dt, $J$ = 14.0, 6.5 Hz, 1H), 5.52 (dd, $J$ = 14.0, 6.2 Hz, 1H), 4.28 (t, $J$ = 5.0 Hz, 1H), 3.85 (dq, $J$ = 7.5, 3.7 Hz, 1H), 3.79 (dd, $J$ = 11.3, 3.9 Hz, 1H), 3.61 (dd, $J$ = 11.3, 3.5 Hz, 1H), 2.79 (t, $J$ = 7.5 Hz, 2H), 2.46 (m, 2H), 2.43 (s, 3H), 2.18 (m, 4H), 1.94 (t, $J$ = 2.6 Hz, 1H), 1.61 (t, $J$ = 7.3 Hz, 2H), 1.50 (q, $J$ = 7.3 Hz, 2H), 1.43–1.33 (m, 2H), 1.30–1.22 (m, 16H). $^{13}$C NMR (CDCl$_3$, 100 MHz, 25°C): δ 174.14, 151.32, 150.86, 144.70, 141.56, 132.33, 130.28, 129.86, 129.30, 122.90, 122.89, 84.97, 74.45, 68.19, 62.41, 54.56, 36.92, 35.35, 33.83, 29.73, 29.71, 29.63, 29.51, 29.42, 29.25, 28.90, 28.62, 25.88, 21.65, 18.53. IR (neat, ATR): 3294, 2920, 2850, 1643, 1603, 1542, 1467, 1377, 1279,

1156, 1104, 1049, 1014, 960, 895, 840, 722. **HRMS (ESI$^+$):** m/z calcd. for $[C_{35}H_{50}N_3O_3]^+$: 560.3847, found: 560.3851 ($[M + H]^+$). **Melting point:** 111°C.

## Acknowledgements

The authors acknowledge financial support from the Deutsche Forschungsgemeinschaft (SFB1032 project B09 to DT and A09 to PS, SFB944 project P14 to JCMH), the Faculty of Biology/Chemistry from the University of Osnabrück (Incentive Award to MK), and the Natural Sciences and Engineering Research Council of Canada (PGS D Scholarship to BW). We thank Dr. Oliver Thorn-Seshold for providing the pulsed LED illumination setup, Prof. Dr. Christian Ungermann for the kind gift of the anti-Vti1p antibody, and CordenPharma Switzerland (Liestal, CH) for providing materials for chemical synthesis.

## Additional information

### Funding

| Funder | Grant reference number | Author |
| --- | --- | --- |
| Deutsche Forschungsgemeinschaft | SFB1032 | Henri G Franquelim<br>Petra Schwille<br>Dirk Trauner<br>James A Frank |
| Natural Sciences and Engineering Research Council of Canada | | Ben Williams |
| Deutsche Forschungsgemeinschaft | SFB944 | Matthijs Kol<br>Joost CM Holthuis |
| university of Osnabrück | Incentive Award | Matthijs Kol |

The funders had no role in study design, data collection and interpretation, or the decision to submit the work for publication.

### Author contributions

Matthijs Kol, Conceptualization, Data curation, Formal analysis, Supervision, Investigation, Visualization, Methodology, Writing—original draft, Writing—review and editing; Ben Williams, Data curation, Formal analysis, Investigation, Methodology, Writing—original draft, Writing—review and editing; Henry Toombs-Ruane, Sergei Korneev, Christian Schroeer, Data curation, Investigation, Methodology; Henri G Franquelim, Data curation, Formal analysis, Investigation, Visualization, Methodology, Writing—original draft, Writing—review and editing; Petra Schwille, Resources, Supervision, Funding acquisition; Dirk Trauner, Conceptualization, Resources, Supervision, Funding acquisition, Writing—original draft, Project administration, Writing—review and editing; Joost CM Holthuis, Conceptualization, Resources, Data curation, Formal analysis, Supervision, Funding acquisition, Visualization, Writing—original draft, Project administration, Writing—review and editing; James A Frank, Conceptualization, Data curation, Formal analysis, Supervision, Validation, Investigation, Visualization, Methodology, Writing—original draft, Project administration, Writing—review and editing

### Author ORCIDs

Matthijs Kol (iD) https://orcid.org/0000-0003-3068-6501
Ben Williams (iD) https://orcid.org/0000-0003-1483-4981
Henri G Franquelim (iD) http://orcid.org/0000-0001-6229-4276
Sergei Korneev (iD) http://orcid.org/0000-0002-1273-5819
Petra Schwille (iD) http://orcid.org/0000-0002-6106-4847
Dirk Trauner (iD) https://orcid.org/0000-0002-6782-6056
Joost CM Holthuis (iD) https://orcid.org/0000-0001-8912-1586
James A Frank (iD) https://orcid.org/0000-0001-6705-2540

Decision letter and Author response
Decision letter https://doi.org/10.7554/eLife.43230.025
Author response https://doi.org/10.7554/eLife.43230.026

## Additional files

### Supplementary files

• Supplementary file 1. Compound Characterization - NMR spectra.
DOI: https://doi.org/10.7554/eLife.43230.022

• Transparent reporting form
DOI: https://doi.org/10.7554/eLife.43230.023

### Data availability

All data generated or analysed during this study are included in the manuscript and supporting source data file. Source data files have been provided for Figures 4-6.

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
