## [Decision Letter]

[Editors’ note: a previous version of this study was rejected after peer review, but the authors submitted for reconsideration. The first decision letter after peer review is shown below.]

Thank you for submitting your work entitled "Optical manipulation of sphingolipid biosynthesis using photoswitchable ceramides" for consideration by *eLife*. Your article has been reviewed by three peer reviewers, and the evaluation has been overseen by a Reviewing Editor and a Senior Editor. The following individuals involved in review of your submission have agreed to reveal their identity: Anant K Menon (Reviewer #3).

Our decision has been reached after consultation between the reviewers. Based on these discussions and the individual reviews below, we regret to inform you that your work will not be considered further for publication in *eLife*.

Tools that can modulate levels of lipids in cells are valuable resources for understanding membrane biology and all three reviewers agreed that the photo-switchable ceramides reported in this work were potentially interesting in this regard. However, the manuscript does not provide a clear vision of the applicability of this tool and it was unclear to the reviewers how these artificial ceramides could provide insights into sphingomyelin biology or other membrane signaling processes. The reviewers had several other concerns and suggestions for improvements (see comments below) and would be interested in reviewing a revised version of the manuscript at a later time point, but only if all their points were fully addressed.

*Reviewer #1:*

In this paper, Frank et al. describe a clever method to modulate the sphingomyelin concentrations of cells. This paper is submitted as a candidate for the "Tools" category of papers for *eLife*. Unlike cholesterol, which can be rapidly depleted by cyclodextrins and rapidly repleted by lipoprotein particles or cholesterol/cyclodextrin complexes, there is no robust way to modulate levels of other lipids. As such, I was excited by the promise of this paper to "optically manipulate sphingolipid biosynthesis".

However, after reading the paper, it is unclear to me whether this method delivers on this promise and even if it did, what could be learned using this method. The authors describe photoswitchable ceramides that can be added to cells and then subsequently converted by plasma membrane SMase2 into sphingomyelin after optical activation with a particular wavelength. The work follows up on their 2016 paper in JACS where they initially describe this concept. However, there are some issues:

The authors claim light-induced induction of domains in membranes (lakes) which can be reversed. However, this reversibility (Figure 3) is not convincing. The claim of light-induced increase in SM synthesis is correct, but this is not technically reversible. The sphingomyelin that has been made (with the extra benzenes) is still there, even when the optical manipulation is reversed. A "pulse" of SM synthesis may be more accurate.

It would be useful to compare the rise in SM levels with the method described here to other methods such as delivery of BSA/sphingomyelin conjugates or delivery of sphingomyelin liposomes (Bierman/Slotte).

A more detailed description of what could be done with this method (in the Discussion) would be helpful to evaluate the utility of this clever method.

*Reviewer #2:*

Frank et al. have recently published papers on photoswitchable azobenzene-lipids, specifically on diacylglycerols and ceramides. They have convincingly demonstrated photoswitching of DAG-activated signalling as well as switching structural parameters of a reconstituted lipid bilayer by a switchable ceramide. In the present manuscript, they extend the previous work on ceramides by addition of a triple bond that can be used for metabolic tracing and explore the usability of the resulting ceramide as a switchable substrate for a major metabolic pathway of ceramides, i.e. conversion by SMS2 to sphingomyelin (SM). As a proof-of-principle study, the present manuscript indeed shows that photoswitching of ceramide as a SMS2-substrate is possible.

Since the manuscript is submitted as a toolbox paper, it should yet also be useful for more widespread applications. At this point I do have major concerns. On the one hand, the new ceramides can be used to photoswitch phase behavior in reconstituted systems, but comparison of the data in Figure 3 with those found in the 2016 paper by the same authors indicate that the simpler ceramides in that study perform much better for this task. On the other hand, some aspects of metabolic conversion of the new ceramides raise questions about the usefulness of the compounds for this purpose. While SMS2 does not discriminate cis- and trans-states for some of the compounds, for some others, most noticeable caCer-1, it is the cis-state that is accepted by the enzyme, but not the trans-state. Natural SM has a fatty acid distribution dominated by 16:0, 18:0 (brain), 20:0, 22:0, 22:1, 24:1 and 24:2. Higher unsaturated species do not appear in relevant amounts. Given that, a substrate with a bulky, irregular shape resembling a polyunsaturated fatty acid (as also demonstrated by the authors in their beautiful DAG study) is hardly representing a biologically relevant species. The fact that the trans-conformation, which in principle should be preferred by the enzyme, is not converted to product raises even more concerns. In vivo (cell culture), the substances are not well metabolized, leaving the ceramides as major species after 1h of incubation. This is in line with previous observations indicating that long-chain ceramides are difficult to deliver to cells and slow in subsequent metabolism. It is unclear which biological properties the produced SM with the very unnatural ceramide backbone would have, and it is, at least to me, also unclear where the application for that would be. More generally, I see the obvious use for switching DAG as a signalling substance, but I cannot see where the application for switching a very artificial ceramide as a substrate for SMS2 really is. If the purpose is just providing or not a metabolic substrate, this can be done (and is regularly being done) by simply applying and withdrawing natural or other labeled precursor lipid to the growth medium in a pulse-chase experiment.

In essence, this is a study with an original idea that is technically well done (with the limitations discussed for Figure 3) but with a very narrow focus lacking broader applicability.

*Reviewer #3:*

The photoswitchable ceramides described in this report are interesting and potentially useful tools, although their precise application is only hinted at by the authors rather than explicitly discussed or demonstrated.

Technical issues:

AFM studies (Figure 3):

The reversibility of the so-called 'lakes' is qualitative at best and it seems that this could depend on the specific image presented – this phenomenon should be scored/quantified somehow.

The L_d_/L_o_ ratio hardly changes during the light regime – how do the authors interpret this? Perhaps a better explanation is necessary or the phenomenon is simply not robust.

SM synthesis rates (Figure 5): there are different rates of synthesis in under blue light depending on the light regime used – sometimes finite (~0.2 units per 10 min), sometimes zero. Why would this be the case? Is the reagent limiting?

Discussion points:

The authors invoke the temporal precision of light activation. Can spatial precision also be useful in this system?

The comment that the light-induced effects are reversible is not true in a practical sense. Although the ability of the ceramide reagents to be used as substrates for SM synthase can be controlled by light, the products formed are permanent for the time of the experiment. Thus control of conditions is not as precise as, for example, in the case of photoswitchable ligands for GPCRs where the same cis-trans isomerization trick can be used to bring a ligand within reach of a binding pocket and then pull it away by changing illumination.

[Editors’ note: what now follows is the decision letter after the authors submitted for further consideration.]

Thank you for resubmitting your work entitled "Optical manipulation of sphingolipid biosynthesis using photo switchable ceramides" for further consideration at *eLife*. Your revised article has been favorably evaluated by three reviewers, and the evaluation has been overseen by a Reviewing Editor and Michael Marletta as the Senior Editor.

The manuscript has been improved but, as outlined below, there are some remaining issues that need to be addressed before a final decision can be reached.

Major points that must be addressed:

1) It is impossible to infer co-localization from Figure 5B alone. Please quantify the extent of overlap between SMS2-V5 and GM130 using appropriate image analysis tools.

2) Irradiation with UV light for 30 sec, as stated in the Materials and methods section might affect the HeLa cells in many ways. As stated in subsection “Optical manipulation of SM biosynthesis in living cells”, 'Cells that did not receive any light treatment and that were kept in the dark throughout the incubation period served as control.', is not sufficient. Cells equally exposed to light but without added probe should serve as an additional control.

3) What is the meaning of a.u. in the SM production from yeast cells in Figure 4? Using standards and a protein determination, the amount of synthesized SM could be provided and normalized to mg of protein. The current presentation is unconventional.

4) Besides studying metabolism, another goal of using dye-clickable lipid probes should be to visualize the tagged lipids in living cells. Has this been attempted? If so, please show or discuss the results, as in case of useful results, this would tremendously increase the usability of the described ceramide analogs. As another discussion point, please address whether possible changes in intracellular localization due to photo-activation could also play a role in the context of the substrate specificity of the synthetases, as one isomer might by shielded compared to the other.

5) The relevance of L_o_/L_d_ partitioning observed in model membranes to membranes of living cells is still under debate. Which cellular membranes do the authors have in mind in which L_o_/L_d_-coexistence has been shown under physiological conditions? The third paragraph of the Discussion which ends with 'Conceivably, isomerization of this compound to trans may lead to a less pronounced clustering in cellular membranes.' should be removed or re-expressed.

---

## [Author Response]

[Editors’ note: a previous version of this study was rejected after peer review, but the authors submitted for reconsideration. The first decision letter after peer review is shown below.]

Reviewer #1:In this paper, Frank et al. describe a clever method to modulate the sphingomyelin concentrations of cells. This paper is submitted as a candidate for the "Tools" category of papers for eLife. Unlike cholesterol, which can be rapidly depleted by cyclodextrins and rapidly repleted by lipoprotein particles or cholesterol/cyclodextrin complexes, there is no robust way to modulate levels of other lipids. As such, I was excited by the promise of this paper to "optically manipulate sphingolipid biosynthesis".However, after reading the paper, it is unclear to me whether this method delivers on this promise and even if it did, what could be learned using this method. The authors describe photoswitchable ceramides that can be added to cells and then subsequently converted by plasma membrane SMase2 into sphingomyelin after optical activation with a particular wavelength. The work follows up on their 2016 paper in JACS where they initially describe this concept.

We acknowledge that the reversibility of light-induced effects on lipid domains in supported lipid bilayers (SLBs) may not have been entirely obvious from the original figure. In our previous study on azobenzene-containing ceramides (ACes; Frank et al., 2016), we observed more pronounced light-induced effects on lipid domains in SLBs containing 19 mol% azolipid than in SLBs containing 11 mol% azolipid. For the original experiments in our current study, SLBs with 11 mol% azolipid were used. Therefore, we have now revised the original figure (formerly Figure 3, now Figure 2) and included new confocal images and high-speed AFM analysis of SLBs containing 19 mol% caCer-3 or caCer-4, in which the reversibility of light-induced effects on lipid domains is more evident. Note that photoisomerisation primarily affects the ratio between L_o_ and L_d_ areas. On isomeration of dark-adapted caCers to *cis* by UV-A light, L_d_ lakes form within the L_o_ phase (Figure 2B). During equilibration, these lakes laterally diffuse toward the L_d_ phase or coalesce into larger lakes in an effort to minimize line tension. On isomerization of caCers to *trans* by blue light, the L_d_ lakes shrink while the surrounding L_o_ area grows “fatter”, essentially occupying the same total area as in the original dark-adapted state. These results are in line with our previous work and can also be appreciated in the new Videos 1 and 2. To demonstrate that the light-induced effects on lipid domains are reversible over multiple cycles, we have also plotted the normalized L_o_ area in both caCer-3 and caCer-4 containing SLBs over time (Figure 2C). These novel data are now described in detail in subsection “caCers enable optical control of ordered lipid domains in supported bilayers”.

However, there are some issues:The authors claim light-induced induction of domains in membranes (lakes) which can be reversed. However, this reversibility (Figure 3) is not convincing. The claim of light-induced increase in SM synthesis is correct, but this is not technically reversible. The sphingomyelin that has been made (with the extra benzenes) is still there, even when the optical manipulation is reversed. A "pulse" of SM synthesis may be more accurate.

The reviewer correctly points out that SM formed from photoswitchable ceramides is essentially permanent, at least within the timespan of the experiment. However, as caCer-1 and caCer-3 serve as light-sensitive substrates, they enable reversible control over the rate of SM production. As demonstrated in Figure 4 (formerly Figure 5) and now explicitly stated in subsection “Optical manipulation of SM biosynthesis in living cells”, this allowed us to generate time-resolved pulses of SM synthesis by light. Moreover, we included new data demonstrating that caCers can also be used to generate pulses in the production of glucosylceramide, the precursor of complex glycosphingolipids (new Figure 6). We have now amended the Abstract, Results and Discussion sections to avoid potential confusion on the utility of caCers as light-sensitive substrates.

It would be useful to compare the rise in SM levels with the method described here to other methods such as delivery of BSA/sphingomyelin conjugates or delivery of sphingomyelin liposomes (Bierman/Slotte).

We thank the reviewer for this remark. It is reasonable to assume that delivery of SM via BSA conjugates or liposomes would result in higher SM levels than what can be achieved with our current method. This is because our approach relies on the delivery of long-chain ceramide analogues, which are considerably more hydrophobic than SM. However, we would like to point out that our method enables superior spatiotemporal resolution for manipulating SM levels and could be used to generate controlled “pulses” of SM in subpopulations of cells in a culture dish, something that is impossible to achieve through addition of SM via BSA conjugates. To circumvent limitations associated with the delivery of hydrophobic compounds and expand opportunities for manipulating the metabolic fate and biological activity of sphingolipids by light, we set out to develop azobenzene-containing sphingosines (caSphs) as a next generation of photoswitchable sphingolipid precursors. We anticipate that these novel compounds will be more readily taken up by cells. As outlined below, we obtained first proof for the suitability of caSphs as light-sensitive substrates to exert optical control over ceramide biosynthesis. These various aspects, which are directly relevant to the general utility of our approach, are now covered in the Discussion (paragraph five).

**Author response image 1. respfig1:** (**A**) Chemical structures of the clickable and azobenzene-containing sphingosines, clickSph and caSph-1. (**B**) UV-A-irradiated or dark-adapted clickSph and caSph-1 were incubated with lysates of yeast cells lacking ceramide synthases (-) or expressing human ceramide synthase CerS5 (+). Metabolic conversion to ceramide was monitored by TLC analysis of lipid extracts click-reacted with Alexa-647-azide. Note that metabolic conversion of caSph-1, unlike that of clickSph, is light sensitive.

A more detailed description of what could be done with this method (in the Discussion) would be helpful to evaluate the utility of this clever method.

This point is well taken. We have now included a new figure (Figure 7) and extended the Discussion to highlight a central concept that was somewhat underexposed in the original manuscript, namely that light-induced changes in lateral packing enable instant and reversible control over both metabolic fate and signaling activity of caCers. As described in paragraph three of the Discussion, a distinct advantage of caCers over caged ceramide analogs is that their ‘bioactivity’ can be switched off by light. This enabled us to generate time-resolved pulses of SM synthesis at the plasma membrane of living cells. Our method offers superior spatiotemporal resolution for manipulating SM levels and could be used to generate controlled SM pulses in subpopulations of cells in a culture dish by patterned illumination, something that is impossible to achieve through traditional pulse-chase approaches. A particularly attractive prospect is to use *cis-trans* isomerization to bring caCers within reach of the binding pocket of a ceramide signaling protein, or to pull them away, thus enabling dynamic control over ceramide-operated signaling pathways (Figure 7). In fact, light-induced changes in lateral packing may be a previously unnoted characteristic of other azobenzene-containing signaling lipids like diacylglycerols, where the *cis*-isomers correspond to the bioactive form. We therefore anticipate that caCers will find use in dissecting the causal roles of ceramides in apoptosis, as tumor suppressors, and as antagonists of insulin signaling. As described in paragraph five of the Discussion, we now also refer to the development of photoswitchable sphingoid bases to bypass limitations associated with the poor aqueous solubility of caCers and expand opportunities for manipulating the metabolic fate and signaling activity of sphingolipids by light.

Reviewer #2:Frank et al. have recently published papers on photoswitchable azobenzene-lipids, specifically on diacylglycerols and ceramides. They have convincingly demonstrated photoswitching of DAG-activated signalling as well as switching structural parameters of a reconstituted lipid bilayer by a switchable ceramide. In the present manuscript, they extend the previous work on ceramides by addition of a triple bond that can be used for metabolic tracing and explore the usability of the resulting ceramide as a switchable substrate for a major metabolic pathway of ceramides, i.e. conversion by SMS2 to sphingomyelin (SM). As a proof-of-principle study, the present manuscript indeed shows that photoswitching of ceramide as a SMS2-substrate is possible.Since the manuscript is submitted as a toolbox paper, it should yet also be useful for more widespread applications. At this point I do have major concerns. On the one hand, the new ceramides can be used to photoswitch phase behavior in reconstituted systems, but comparison of the data in Figure 3 with those found in the 2016 paper by the same authors indicate that the simpler ceramides in that study perform much better for this task.

We thank the reviewer for this remark, which has now been addressed as outlined in our response to comment #1 of reviewer #1. In brief, we included new data demonstrating that the light-induced effects on lipid domains in model bilayers containing caCer-3 or caCer-4 are reversible over multiple cycles (new Figure 2C), very similar to what we reported in Frank et al., 2016, for model bilayers containing the simpler ceramides.

On the other hand, some aspects of metabolic conversion of the new ceramides raise questions about the usefulness of the compounds for this purpose. While SMS2 does not discriminate cis- and trans-states for some of the compounds, for some others, most noticeable caCer-1, it is the cis-state that is accepted by the enzyme, but not the trans-state. Natural SM has a fatty acid distribution dominated by 16:0, 18:0 (brain), 20:0, 22:0, 22:1, 24:1 and 24:2. Higher unsaturated species do not appear in relevant amounts. Given that, a substrate with a bulky, irregular shape resembling a polyunsaturated fatty acid (as also demonstrated by the authors in their beautiful DAG study) is hardly representing a biologically relevant species. The fact that the trans-conformation, which in principle should be preferred by the enzyme, is not converted to product raises even more concerns.

It is indeed remarkable that the *cis*-isoforms of caCers (notably caCer-1 and caCer-3) are better substrates for SMS2 than the *trans*-isoforms, as the latter resemble native ceramides more closely. We have now included new data showing that glucosylceramide synthase, a sphingolipid biosynthetic enzyme structurally unrelated to SMS2, also preferentially accepts the *cis*-isoforms as substrates for the production of glucosylceramide (GlcCer; new Figure 6). Although highly saturated ceramides are the most common ceramide species in cells, most literature indicates that SM and GlcCer synthases are relatively tolerant towards structural deviations in the ceramide backbone (Koval and Pagano, 1991; Huitema et al., 2004; Haberkant et al., 2016). Combined with the atomic force microscopy data (new Figure 2), these results support the idea that the marked differences in metabolic conversion between *trans*- and *cis*-isoforms of caCers are due to light-induced alterations in their lateral packing. Thus, we postulate that blue light triggers self-assembly of caCers into tightly packed clusters driven by intermolecular stacking of the flat *trans*-azobenzene groups, thereby reducing their availability for metabolic conversion by SM and GlcCer synthases. In contrast, UV-A irradiation causes caCers to disperse in the membrane as the bent *cis*-azobenzene groups disrupt intermolecular stacking, making them more accessible for metabolic conversion. As illustrated by the model in new Figure 7B, these light-induced changes in lateral packing would enable instant and reversible control over metabolic fate and signaling activity of caCers and other azobenzene-containing lipids. Indeed, our previous work on photoswitchable DAGs revealed that the *cis*-isomer of these compounds correspond to the bioactive form that promotes translocation of protein kinase C toward the plasma membrane (Frank et al., 2016). Moreover, the model is supported by biophysical studies demonstrating that the *trans*-isomers of photoswitchable phosphatidylcholines are more tightly packed in the membrane than the *cis*-isomers (Pernpeintner et al., 2017, *Langmuir* 33, 4083), and helps explain our recent finding that the *cis*-isoform of a photoswitchable sphingosine is the preferred substrate for ceramide biosynthesis (see our response to comment #3 of reviewer #1). These considerations, which are directly relevant to the general utility of our approach, are now covered in the Discussion (paragraph two to four).

In vivo *(cell culture), the substances are not well metabolized, leaving the ceramides as major species after 1h of incubation. This is in line with previous observations indicating that long-chain ceramides are difficult to deliver to cells and slow in subsequent metabolism.*

This is a valid point. To circumvent limitations associated with the relatively inefficient uptake of caCers by cells and to expand opportunities for manipulating the metabolic fate and biological activity of sphingolipids by light, we set out to develop clickable, azobenzene-containing sphingosines (caSphs) as novel photoswitchable sphingolipid precursors. As outlined in our response to comment #3 of reviewer #1, we obtained first experimental proof for the suitability of caSphs as light-sensitive substrates to exert optical control over ceramide biosynthesis. This issue, which is also directly relevant to the general utility of our approach, is now covered in the Discussion (paragraph five).

It is unclear which biological properties the produced SM with the very unnatural ceramide backbone would have, and it is, at least to me, also unclear where the application for that would be. More generally, I see the obvious use for switching DAG as a signalling substance, but I cannot see where the application for switching a very artificial ceramide as a substrate for SMS2 really is. If the purpose is just providing or not a metabolic substrate, this can be done (and is regularly being done) by simply applying and withdrawing natural or other labeled precursor lipid to the growth medium in a pulse-chase experiment.In essence, this is a study with an original idea that is technically well done (with the limitations discussed for Figure 3) but with a very narrow focus lacking broader applicability.

We acknowledge that the general applicability of our method was underexposed in the original manuscript. As outlined in our response to comment #4 or reviewer #1, we have now included a new figure (Figure 7) and extended the Discussion to provide a more detailed description of what could be done with our method and highlight its utility.

Reviewer #3:The photoswitchable ceramides described in this report are interesting and potentially useful tools, although their precise application is only hinted at by the authors rather than explicitly discussed or demonstrated.

We acknowledge that this point was underexposed in the original version of the manuscript. To highlight the broader applicability of caCers and other photoswitchable lipids, we revised the Discussion and also included a new figure (Figure 7). Details of these revisions are outlined in our responses to comment #4 of reviewer #1 and comment #2 of reviewer #2.

Technical issues:AFM studies (Figure 3):The reversibility of the so-called 'lakes' is qualitative at best and it seems that this could depend on the specific image presented – this phenomenon should be scored/quantified somehow.The L_d_/L_o_ ratio hardly changes during the light regime – how do the authors interpret this? Perhaps a better explanation is necessary or the phenomenon is simply not robust.

We thank the reviewer for this remark, which has now been addressed as outlined in our response to comment #1 of reviewer #1. In brief, we have included new data demonstrating that the light-induced effects on lipid domains in model bilayers containing caCer-3 or caCer-4 (changes in L_o_/L_d_ ratio) are reversible over multiple cycles (new Figure 2C).

SM synthesis rates (Figure 5): there are different rates of synthesis in under blue light depending on the light regime used – sometimes finite (~0.2 units per 10 min), sometimes zero. Why would this be the case? Is the reagent limiting?

The reviewer rightly notes that there is some variability in the rate of conversion in the blue regime, depending on whether conversion is measured in an early or later time interval. As a first measure to prevent the enzyme reaction from reaching saturation, we chose the shortest time regime that was still compatible with sample manipulation as detailed in the Materials and methods section. Judged from the cCer measurements, the enzyme reaction is indeed linear over 0-30 min. However, for the caCers there is a trend to lower conversion rates with increasing time, indicative of a slight saturation of the enzyme reaction. Under all conditions, the amount of SM formed was only a few percent of the total amount of available ceramide substrate. Click reactions were performed with Alexa-azide in molar excess over total alkynes. Therefore, these reagents unlikely become limiting during the experiment. The rate of SM production also relies on the availability of the phosphocholine head group donor PC. While PC is generally considered an abundant membrane component, yeast harbors Golgi- and PM-resident flippases that translocate PC from the exoplasmic to the cytosolic leaflet (Alder-Baerens et al., 2006, *Mol Biol Cell* 17, 1632). It is conceivable that these flippases render the pool of exoplasmic PC available for SMS2-mediated SM production limiting. However, experimental validation of this possibility goes beyond the scope of our current study. In spite of a partial saturation effect, we consistently observed that for caCer-1 and caCer-3 the rates of SM synthesis under UV-A light were significantly higher than those under blue light.

Discussion points:The authors invoke the temporal precision of light activation. Can spatial precision also be useful in this system?

As now described in the Discussion, our method offers superior spatiotemporal resolution for manipulating sphingolipid levels and could be combined with patterned illumination to generate controlled sphingolipid pulses in subpopulations of cells or subcellular organelles, something that is impossible to achieve through traditional pulse-chase approaches. Moreover, *cis-trans* isomerization of caCers could be used alongside patterned illumination to dissect the causal roles of ceramides in apoptosis, as tumor suppressors, and as antagonists of insulin signaling (Discussion paragraph three).

The comment that the light-induced effects are reversible is not true in a practical sense. Although the ability of the ceramide reagents to be used as substrates for SM synthase can be controlled by light, the products formed are permanent for the time of the experiment. Thus control of conditions is not as precise as, for example, in the case of photoswitchable ligands for GPCRs where the same cis-trans isomerization trick can be used to bring a ligand within reach of a binding pocket and then pull it away by changing illumination.

We acknowledge that SM formed from caCers is essentially permanent. However, as caCer-1 and caCer-3 serve as light-sensitive substrates, they enable reversible control over the rate of SM production. As demonstrated in Figure 4 (formerly Figure 5) and now explicitly stated in paragraph two of subsection “Optical manipulation of SM biosynthesis in living cells”, this allowed us to generate pulses of SM synthesis by light. Moreover, we have included new data indicating that glucosylceramide synthase, an enzyme structurally unrelated to SM synthase, also preferentially accepts *cis*- over *trans*-isomers as substrates for glucosylceramide biosynthesis (new Figure 6). As illustrated in the model in new Figure 7B, we postulate that the marked differences in metabolic conversion between *trans*- and *cis*-isoforms of caCers are due to light-induced alterations in their lateral packing. In line with the atomic force microscopy data (new Figure 2), blue light would trigger self-assembly of caCers into tightly packed clusters driven by intermolecular stacking of the flat *trans*-azobenzene groups, thereby reducing their availability for metabolic conversion by sphingolipid biosynthetic enzymes. UV-A irradiation, on the other hand, causes caCers to disperse in the membrane, as the bent *cis*-azobenzene groups would disrupt intermolecular stacking. This, in turn, would make these reagents more accessible for metabolic conversion. As indicated in Figure 7B and outlined in our response to comment #2 of reviewer #2, we believe that essentially the same physicochemical principles may apply in the case of photoswitchable ligands for GPCRs, where *cis-trans* isomerization can be used to bring the ligand within reach of a binding pocket or pull it away. These considerations, which are directly relevant to the general utility of our approach, are now covered in the Discussion (paragraph four).

[Editors’ note: what now follows is the decision letter after the authors submitted for further consideration.]

The manuscript has been improved but, as outlined below, there are some remaining issues that need to be addressed before a final decision can be reached.Major points that must be addressed:1) It is impossible to infer co-localization from Figure 5B alone. Please quantify the extent of overlap between SMS2-V5 and GM130 using appropriate image analysis tools.

As requested, we have further developed our image analysis to better infer the partial co-localization of SMS2-V5 and the Golgi marker GM130. To this end, we pseudo-colored the α-V5 and α-GM130 channels green and magenta, respectively, alongside addition of a merge to demonstrate overlap (see updated Figure 5B). Moreover, we included intensity plots to show the level of co-localization between SMS2-V5 and GM130 in a more quantitative manner. Our fluorescence microscopy analysis indicates that SMS2-V5 is predominantly localized to the plasma membrane, and to a lesser extent to the Golgi apparatus. This is in line with the results obtained from subcellular membrane fractionation studies [Tafesse et al., 2016, J. Biol. Chem. 282, 17537-17547; Ref. 35].

2) Irradiation with UV light for 30 sec, as stated in the Materials and methods section might affect the HeLa cells in many ways. As stated in subsection “Optical manipulation of SM biosynthesis in living cells”, 'Cells that did not receive any light treatment and that were kept in the dark throughout the incubation period served as control.', is not sufficient. Cells equally exposed to light but without added probe should serve as an additional control.

We thank the reviewer for pointing out this ambiguity in our wording. In fact, the “no irradiation” condition was used as a comparison to obtain a baseline amount of SM that was formed from incubation of SMS2 overexpressing HeLa cells with *trans*-caCer only. To demonstrate dynamic photoswitching over the course of the experiment in living cells (now seen in Figure 5I), the amount of SM formed in the untreated cells is similar to that formed by cells that received the UV-A→blue treatment. This is expected, as this irradiation protocol generates trans-caCer-1 for the majority of the incubation. In contrast, when cells received the blue→UV-A treatment (middle bars), cis-caCer-1 is present throughout the majority of the incubation, and consequently more SM is generated under these irradiation conditions.

For clarity, we have modified the described sentence to read:

“Cells that did not receive any light treatment and that were kept in the dark throughout the incubation period served as baseline for the conversion of the trans-isomer only.”

We cannot perform the suggested control experiment, as light-treated cells lacking the probe would not possess any alkyne functionalized lipids, thus there would be no signal to readout on the TLC. In our opinion, HeLa cells incubated with the non-switchable cCer serve as a suitable control for the effects of UV-A (or blue) illumination on the cells.

For clarification, we adapted the text to read:

“In contrast, production of SM from cCer was not affected by light treatment, demonstrating that UV-A irradiation itself did not affect SM-production in HeLa cells”

3) What is the meaning of a.u. in the SM production from yeast cells in Figure 4? Using standards and a protein determination, the amount of synthesized SM could be provided and normalized to mg of protein. The current presentation is unconventional.

As described in the Materials and methods section, the relative SM/(SM+Cer) signal was normalized to the maximal fractional SM obtained within the 30 min incubation. To make this more intuitive, we changed the y-axis labeling of Figure 4A to read *% of maximum* instead of *a.u.* We furthermore note that the amount of protein in each of the samples across all time points in Figure 4A was equal, and that therefore normalization to the amount of total protein in the crude yeast lysate was already taken into account.

4) Besides studying metabolism, another goal of using dye-clickable lipid probes should be to visualize the tagged lipids in living cells. Has this been attempted? If so, please show or discuss the results, as in case of useful results, this would tremendously increase the usability of the described ceramide analogs.

We agree that this is an exciting application of our tools. As a prelude to imaging experiments in living cells, we have set out to incorporate fluorophore-tagged photoswitchable ceramides in supported lipid bilayers and examine their conformation-dependent behavior using single-molecule tracking. However, given the sophisticated and time-consuming nature of these experiments and the already broad scope of the present manuscript, these studies remain out of the scope of the current revision.

As another discussion point, please address whether possible changes in intracellular localization due to photo-activation could also play a role in the context of the substrate specificity of the synthetases, as one isomer might by shielded compared to the other.

Please note that in the experiments on living cells (Figure 5D-I), we mainly monitor the impact of photoswitching on metabolic conversion of externally added caCers by SMS2-V5, an enzyme predominantly located at the plasma membrane. In addition, the data obtained with cell-free produced enzymes (Figures 3D-F and 6G) clearly indicate that metabolic conversion per se is affected by light treatment. Consequently, while one could imagine that intracellular transport of caCers in living cells may also be influenced by photoswitching, this scenario is largely irrelevant for the data presented in our paper.

5) The relevance of L_o_/L_d_ partitioning observed in model membranes to membranes of living cells is still under debate. Which cellular membranes do the authors have in mind in which L_o_/L_d_-coexistence has been shown under physiological conditions? The third paragraph of the Discussion which ends with 'Conceivably, isomerization of this compound to trans may lead to a less pronounced clustering in cellular membranes.' should be removed or re-expressed.

Although we appreciate the ongoing debate on the existence of lipid rafts (i.e. segregated L_o_/L_d_domains) within living cells, at no point in this manuscript do we propose that the existence of lipid rafts drives these observed effects in living cells. To avoid potential confusion regarding this matter, we adapted the third paragraph of the Discussion as follows:

“Based on our findings in raft-mimicking SLBs, we hypothesize that the impact of cis-trans isomerization on the metabolic fate of caCers may be caused by light-induced alterations in their lateral packing (Figure 7B). According to this model, trans-caCers pack more tightly within the membrane due to favorable intermolecular stacking interactions among the flat trans-azobenzene groups, thereby reducing their availability for enzymatic conversion. On isomerization to cis, the kinked caCer molecules become dispersed throughout the membrane, making them more accessible for enzymatic conversion. This idea is supported by our finding that trans-caCers preferentially localize in liquid-ordered (L_o_) domains in model bilayers (Figure 2, Frank et al., 2016). caCer-4 forms an exception, as its isomerization to trans promotes association with L_o_ domains without having an obvious impact on SM biosynthesis. While both caCer-3 and caCer-4 have azobenzene-containing sphingoid bases, caCer-4 is unique among all caCers in that its azobenzene photoswitch is positioned close to the end of the carbon chain. Conceivably, isomerization of this compound to trans may lead to a less pronounced clustering in cellular membranes.”